# Trends in Photothermal Nanostructures for Antimicrobial Applications

**DOI:** 10.3390/ijms24119375

**Published:** 2023-05-27

**Authors:** Violeta Dediu, Jana Ghitman, Gratiela Gradisteanu Pircalabioru, Kiat Hwa Chan, Florina Silvia Iliescu, Ciprian Iliescu

**Affiliations:** 1National Research and Development Institute in Microtechnologies—IMT Bucharest, 126A Erou Iancu Nicolae Street, 077190 Voluntari, Romania; violeta.dediu@imt.ro; 2eBio-hub Research-Center, University “Politehnica” of Bucharest, 6 Iuliu Maniu Boulevard, Campus Building, 061344 Bucharest, Romania; jana.ghitman@upb.ro (J.G.); ggradisteanu@upb.ro (G.G.P.); 3Advanced Polymer Materials Group, University Politehnica of Bucharest, 1-7 Gh. Polizu Street, 011061 Bucharest, Romania; 4Academy of Romanian Scientists, 54 Splaiul Independentei, 050094 Bucharest, Romania; 5Research Institute of University of Bucharest, University of Bucharest, 050095 Bucharest, Romania; 6Division of Science, Yale-NUS College, 16 College Avenue West, Singapore 138527, Singapore; kiathwa.chan@yale-nus.edu.sg; 7NUS College, National University of Singapore, 18 College Avenue East, Singapore 138593, Singapore

**Keywords:** photothermal antimicrobials, antibacterial mechanisms, antibiofilm, wound healing

## Abstract

The rapid development of antimicrobial resistance due to broad antibiotic utilisation in the healthcare and food industries and the non-availability of novel antibiotics represents one of the most critical public health issues worldwide. Current advances in nanotechnology allow new materials to address drug-resistant bacterial infections in specific, focused, and biologically safe ways. The unique physicochemical properties, biocompatibility, and wide range of adaptability of nanomaterials that exhibit photothermal capability can be employed to develop the next generation of photothermally induced controllable hyperthermia as antibacterial nanoplatforms. Here, we review the current state of the art in different functional classes of photothermal antibacterial nanomaterials and strategies to optimise antimicrobial efficiency. The recent achievements and trends in developing photothermally active nanostructures, including plasmonic metals, semiconductors, and carbon-based and organic photothermal polymers, and antibacterial mechanisms of action, including anti-multidrug-resistant bacteria and biofilm removal, will be discussed. Insights into the mechanisms of the photothermal effect and various factors influencing photothermal antimicrobial performance, emphasising the structure–performance relationship, are discussed. We will examine the photothermal agents’ functionalisation for specific bacteria, the effects of the near-infrared light irradiation spectrum, and active photothermal materials for multimodal synergistic-based therapies to minimise side effects and maintain low costs. The most relevant applications are presented, such as antibiofilm formation, biofilm penetration or ablation, and nanomaterial-based infected wound therapy. Practical antibacterial applications employing photothermal antimicrobial agents, alone or in synergistic combination with other nanomaterials, are considered. Existing challenges and limitations in photothermal antimicrobial therapy and future perspectives are presented from the structural, functional, safety, and clinical potential points of view.

## 1. Introduction

Bacterial infections and related antimicrobial resistance are under surveillance by the World Health Organization (WHO) and Centers for Disease Control and Prevention (CDC) worldwide [1], in all healthcare sectors and agriculture [2], due to the increased morbidity and mortality that they cause [3]. Increasing antimicrobial resistance is one of the top ten “global public health threats facing humanity” [4,5], thus impacting individuals at every level of life: personal, professional, and societal [6,7]. Bacteria harm humans directly through their resistance to commonly used antibiotics [8] or the resulting severe adverse effects induced by second- and third-line treatments of nosocomial infections, primarily antibiotic-resistant infections [4,6,9,10,11,12,13]. One of the most pressing challenges is to define new antibacterial materials and strategies with high efficiency, safety, and convenience [14], knowing that traditional drugs or methods failed due to drug resistance [15,16]. Nanotechnology-based delivery systems and engineered nanoparticles have been developed as alternative “nanoantibiotics” [17]. Nanoparticles (NPs) have been demonstrated to be the most effective method to address multidrug-resistant bacteria since they not only act as transporters for natural antibiotics and antimicrobials but also actively combat bacteria. Inorganic NPs (e.g., silver (Ag) [18,19,20,21], zinc oxide (ZnO) [22,23,24], gold (Au) [25,26,27], titanium oxide TiO_2_ [28], copper (Cu) [29], copper oxide (CuO) [30], nickel (Ni) [31], and selenium (Se) [32]) and natural and synthetic organic NPs (e.g., liposomes, polymeric nanoparticles, micelles, and ferritin) [33,34,35] can be used alone or as nanocarriers for therapeutic molecules (e.g., liposomes, polymeric NPs, and dendrimers) [36,37,38]. Hybrid NPs combine organic and inorganic NPs in the same composite system [39]. Nanomaterials have been used as antibacterial agents, and dynamic therapies have been designed for better efficacy by increasing drugs’ bioavailability, ensuring their targeted distribution, and decreasing their toxicity [40,41,42]. Notably, the nanomaterials’ physical and chemical properties (dimensions under 100 nm, morphology, crystal structure, defect state, surface energy, surface potential) can be tuned to meet the requirements of specific applications [43,44]. Moreover, stimulus-based, tuneable, noninvasive approaches include dynamic therapy, which employs various stimuli, such as thermal (photothermal therapy) [45], chemical (chemodynamic therapy—CDT) or electrical [46] (photodynamic therapy—PDT), immunotherapy [47], and gene therapy [48], each of them with their advantages and limitations [17,49,50]. In the case of localised surface plasmon resonance (LSPR) by photothermally active nanomaterials (photothermal agents—PTAs), the absorbed energy (near-infrared (NIR) light, 700–1300 nm) is released as heat (hyperthermia up to 90 °C). It alters the membrane, inactivates proteins, and releases intracellular material to fine-tune the photothermal ablation of bacteria in situ but can damage the surrounding cells [49]. The photothermal effect, occurring in the range of the biological transparency window [51], allows the deep penetration of light (up to 1cm) into infected tissue [52], avoids mutations in the targeted bacteria, and kills planktonic multidrug-resistant (MDR) microorganisms and biofilm. However, the short- and long-term biological concerns restricting PTT applications requires functionalising PTAs for specific bacteria and moving towards lower-energy (i.e., NIR-II 1000–1700 nm and NIR-III 1800–2100 nm) or active photothermal materials for multimodal synergistic-based therapies to minimise side effects and maintain low costs (e.g., PTT-PDT, PTT-CDT, PTT-photocatalytic, PTT-immunotherapy, and PTT-catalytic) [53,54].

Here, we review the recent achievements and current trends in developing photothermally active nanostructures, including plasmonic metals, semiconductors, and carbon-based and organic photothermal polymers, and antibacterial mechanisms of action (MOA), including anti-MDR bacteria and biofilm removal. In addition, new non-conventional photothermal-based antimicrobial systems with remarkable synergistic effects are presented. The most relevant PTA applications are reviewed, such as antibiofilm formation, biofilm penetration or ablation, and nanomaterial-based infected wound therapy. We provide an overview of the strengths, limitations, and general challenges of photothermal treatment using nanomaterials to highlight research directions.

## 2. Photothermal Antimicrobial Mechanism

When subjected to heating at temperatures above 45 °C, the viability of most bacteria is altered (Figure 1). Depending on the PTAs used, different mechanisms of photothermal conversion can occur: localised surface plasmon resonance in metals, electron–hole generation and relaxation in semiconductors, and HOMO (highest occupied molecular orbital)–LUMO (lowest unoccupied molecular orbital) excitation and lattice vibration of molecules [55]. 

Metal NPs’ photothermal antibacterial action relies on the absorption of visible light radiation with plasmon formation (the collective oscillation of conduction electrons), followed by a high-speed transfer of the generated heat to the surrounding bacteria, inducing cell/bacteria death. The outcomes of photothermal bacterial ablation depend on the NPs’ size, shape, and dielectric constant and the surrounding materials’ permittivity, the incident light wavelength, and the light intensity. These parameters can be tuned for the optimisation of photothermal interactions [56]. Semiconductors act through the generation and relaxation of electron–hole pairs after irradiation by incident light with an energy similar to the bandgap [57]. Carbon- and polymer-based materials generate a photothermal effect through the lattice vibration of molecules. Upon illumination with light energy, the excited electron goes from the ground state (LUMO) to a higher-energy orbital (HOMO). Next, electron–phonon coupling enables the relaxation from higher excitation states to lower-energy states. Hence, the energy gained is conveyed from excited electrons to vibrational modes within the atomic lattices, triggering an increase in temperature [57,58]. The photothermal action cannot be isolated from other mechanisms of antimicrobial action, such as nanoperforation, the destruction of membrane stability, biomolecule binding, and oxidative damage [59]. These interactions between NPs and bacteria depend on many factors, such as the NP’s surface chemistry, charge, and hydrophobicity. The photothermal action mainly depends on the dielectric constants of PTA nanoparticles and the surrounding materials.

## 3. Photothermal Antimicrobial Agents

Photothermal antimicrobial agents are nanostructures with significant absorption in the NIR region. The selection of the most suitable PTAs is critical for the success of antimicrobial therapy, considering that they must comply with specific requirements: physical, chemical, or biological (e.g., photostability, suitable size, and good biocompatibility). Light-to-heat conversion can be optimised by tuning the nanoparticle size, shape, and composition. PTAs can be categorised according to their chemical composition into plasmonic metals, semiconductors, and carbon- and polymer-based materials. From a structural point of view, various nanostructures have been tested as photothermal antimicrobial agents. There are 0-dimensional (e.g., quantum dots such as graphene, carbon, and black phosphorus), 1D (CNT, AuNR), 2D (black phosphorus nanosheets, boron nitride, and graphitic carbon nitride, MXenes), and 3D (covalent organic framework and metal–organic framework) materials. Their characteristics and photothermal antimicrobial performance will be discussed in this section, divided into inorganic- and organic-based photothermal antimicrobial materials.

### 3.1. Inorganic-Based Photothermal Antimicrobial Agents

Metals (such as Au, Cu, Pd, Bi), semiconductor oxides (WO_3_, Fe_3_O_4_), semiconductor chalcogenides or dichalcogenides (CuS, MoS_2_, NiS_2_, SnSe, CuSe), metalloids (B), and nonmetals (C, P) are inorganic nanostructures that absorb energy strongly in the NIR region. Table 1 presents demonstrative metal-based PTAs and other inorganic nanomaterials and their antibacterial activities. 

#### 3.1.1. Plasmonic-Metal-Based PTAs

Metal-based NPs are at the forefront of fighting bacteria as “light-directed nanoheaters” due to their substantial light-to-heat conversion efficiency. Recent achievements focused on developing plasmonic metals have boosted their use in antimicrobial applications. Plasmonic metal NPs can exert effective and targeted antibacterial activity against a broad spectrum of bacterial strains. 

The most used plasmonic metal for bacterial photothermolysis is gold (Au) in different shapes, sizes, and structures. Nano-gold (nanorods, nanostars, nanobipyramids, nanowires, nanoworms, nanoflowers) present fascinating localised surface plasmonic resonance (LSPR) properties, while their chemical inertness makes gold nanostructures suitable for bacterial ablation through photoinduced hyperthermia. Au NPs have been widely adopted for biological applications due to their easy nanoscale fabrication and high oxidation and degeneration resistance [60]. Controlling the shape and size of Au nanostructures enables the optical tuning of LSPR activity at different light wavelengths, from the visible to the NIR region [61,62]. Using Au NPs for bacterial photothermal ablation has been an evolving application, from the first in vitro study using bioconjugated Au plasmonic NPs under laser light [63] to the proof of principle of PTT biofilm removal [64] and the involvement of Au nanostructures in PTT bacterial ablation [65,66]. 

Au nanorods (Au NRs) have been the most utilised because of their longitudinal surface plasmon resonances under NIR laser illumination. In the case of Au NRs, the photothermal bacterial ablation efficiency depends on the Au nanocrystals’ shape, size, overall structure, and, most importantly, facets. Yougbare et al. compared Au NRs with the (200) plane and Au nanobipyramids (Au NBPs) with the (111) plane and found that the photothermal activity of Au NBPs (111) was better against *E. coli* due to the easily desorbed water on the Au NBP (111) surface for PTT hyperthermia [67]. However, pulsed laser irradiation induces structural damage and shape modification to Au NRs [68,69]. Imposing an additional treatment improves their structural stability under laser treatment [40]. For instance, covering anisotropic Au nanomaterials with polydopamine to synthesise Au nanoworms resulted in stability after seven cycles of laser irradiation, efficient antibacterial activity, and good biocompatibility [70]. Furthermore, various species of ligands can be anchored on the AuNP surface for the surface modification of AuNPs to improve the photothermal antibacterial treatment. For example, Hu et al. [71] coated Au NPs with a pH-responsive mixed-charged zwitterionic layer for good dispersion in the biological environment (pH ~7.4), excellent adherence to negatively charged methicillin-resistant *S. aureus* bacterial surfaces (pH ~5.5), and increased PTT performance. Similarly, Au NRs were functionalised for pH-responsive surface charge transition activities using polymethacrylate with pendant carboxyl betaine groups [72]. The hydrophobic/hydrophobic functionalisation of Au NRs substantially improved the antimicrobial efficiency, promoting membrane disintegration. Hydrophilic functionalised polyethylene glycol (PEG)-Au NRs and hydrophobic functionalised polystyrene (PS)-Au NRs showed efficient bactericidal effects on *S. aureus* and *Propionibacterium acnes* (*P. acnes*) strains: the viable bacterial count was reduced from ≤85% to ≥99.99% after exposure to NIR [73]. Other attempts to functionalise Au NRs consisted of (1) conjugating Au NRs with poly(2-lactobionamidoethyl methacrylate) and poly(2-fucose ethyl methacrylate) to specifically block the bacterial LecA and LecB lectins of *P. aeruginosa*, which mediate biofilm formation [74]. In vivo experiments showed a fast temperature increase up to 60 °C and the removal of most bacteria from the infected tissue. Protease (bromelain)-conjugated Au NRs were also used to achieve rapid biofilm thermal degradation and the advanced removal of exotoxins and auto-induced peptides. The enhanced enzymatic activity of bromelain against Gram-positive and Gram-negative bacteria upon NIR laser irradiation was observed. It was regulated within 30–60 °C by adjusting the laser power [75]. A peptide and a neuropeptide were used to functionalise the surface of Au NRs through electrostatic interactions for targeted methicillin-resistant *S. aureus* and *E. coli* binding and had higher bactericidal activity than unconjugated Au NRs [76]. The functionalised photothermal materials were also stable for up to four cycles of NIR laser irradiation. 

Silver (Ag), like Au, exhibits strong plasmonic properties in the visible region, where biological tissues absorb. Merkl et al. [77] obtained Ag plasmonic fractal-like nanoaggregates with tuneable extinction from visible to NIR wavelengths. Using SiO_2_ during the flame synthesis of the spherical Ag NPs, a dielectric spacer was created between plasmonic Ag NPs to tune plasmonic coupling, prevent any potential dissolution of Ag, and inhibit nanostructures’ sintering or restructuring. The resulting nanomaterial was incorporated into a polymer layer and used as photothermal coatings on medical devices. Continuous laser irradiation at 808 nm completely eradicated *E. coli* biofilms after 5 min and *S. aureus* after 10 min. Interestingly, as Ag dissolves in biological media and hyperthermia accelerates the process and consumes the photothermal agent during PTT, Ag NP-embedded hydrogel was developed to release Ag^+^, acting as a PTA for antimicrobial PTT under an NIR laser [78]. Thus, nanostructures with plasmonic properties should be engineeringly changed to absorb in the NIR regions and be used in wound healing [79,80,81].

Copper (Cu) alone is highly oxidised. Therefore, nanoCu can act as a nanoenzyme [82] with high photothermal conversion efficiency as it undergoes Fenton-like reactions in a wide pH range. Notably, the atomically dispersed Cu ensured the photothermal properties and significantly improved the catalytic performance of Cu single-atom sites/N doped porous carbon (Cu SASs/NPC), which showed 100% antibacterial efficiency against *E. coli* and *MRSA* through photothermal–catalytic antibacterial treatment [83]. Furthermore, Cu SASs/NPC demonstrated good photothermal stability due to the structural configuration.

Palladium (Pd), used for photothermal cancer therapy, exhibits good photothermal stability and high optical extinction coefficients, which are useful in bacterial photothermolysis. Recently, biocompatible PdNPs prepared via *Bacillus megaterium* Y-4 and biologically reduced and ultrasonically treated [84] presented improved photothermal conversion and bacterial ablation at low doses through improved absorption in the NIR region. 

Bismuth (Bi)-based nanomaterials with a bandgap of less than 1.53 eV can absorb in the NIR range. Pristine Bi, Bi-based compound nanomaterials, and composites also exhibited antibacterial PTT and PDT capabilities [85], thus having the potential to treat bacterial infections. However, Bi oxidises during irradiation, so strategies must be implemented to prevent this phenomenon and increase the Bi biomedical applications. 

Bimetallic plasmonics. The chemical stability of Ag can be improved by debasing it with Au to expand its overall functionality. Ag/Au bimetallic NPs were synthesised onto a jellyfish-based scaffold. This antibacterial material can actively and spontaneously reduce Ag and Au ions and form NPs directly on the nanofibres’ surface due to Q-mucin glycoproteins’ presence in nanofibres [86]. The heat generated by small plasmonic NPs is more significant than the heat from the bigger and scattered NPs. The resulting materials were proven to exert combined actions against bacterial biofilm: disrupting/removing bacterial colonies and mature biofilms and preventing their regrowth. Another way to preserve the stable shapes of Ag NPs is to frame them with a more stable metal. Zhang et al. [79] proposed another type of architecture of AuAg yolk–shell cubic nanoframes, with the nanosphere as the core and the cubic nanoframe as the outer shell. The existence of a void between the core and shell parts comes with some advantages, such as multiple reflections of the incident light between the shell and core parts and an extensive electromagnetic field interaction between these unconnected parts. The material depolarises the bacterial membrane and affects the membrane potential, and NIR laser exposure further increases the initial effect. TEM images of exposed *MRSA* showed the leakage of intracellular substances. A Pd–Cu nanoalloy, in combination with amoxicillin and encapsulated in zeolitic imidazolate framework-8, formed a complex antimicrobial system [87]. The photothermal nanoalloy significantly stimulates drug release, has good biocompatibility, and has a significant antibacterial effect on planktonic bacteria and their biofilms. 

#### 3.1.2. Metallic-Compound-Based PTAs

Metal sulphides, oxides, selenides, and carbides, with a lower cost than noble metals, are also used as PTAs due to their large surface areas and facile surface modification. The nanomaterials have high photothermal conversion efficiency due to the large bandgap. Some representative metallic compounds used as photothermal agents and their antimicrobial applications are presented in Table 1.

Copper sulphide (CuS) NPs transform light into heat due to the d–d transition of Cu^2+^, and the maximum absorption peak cannot be shifted by changing the particle morphology. In practice, an NIR laser at 980 nm is used because some Cu-based nanomaterials need a high-powered NIR laser at 808 nm. The antimicrobial performance of CuS nanosheets via synergistic photothermal and photodynamic mechanisms depends on the sulphur vacancies’ (Vs) concentration [88]. In the case of CuS, CuS nanosheets with the highest Vs concentration achieved bactericidal rates of 99.9% against *Bacillus subtilis* and *E. coli* bacteria under 808 nm laser irradiation. The photothermal conversion efficiency was 41.8%. Similar results were obtained for other defect-rich CuS [89]. Recently, Chan et al. proposed a multifunctional platform (HNTs@CuS@PDA-Lys) to treat bacterial infections by synergistic lysozyme (Lys)-photothermal therapy [90]. The complex platform includes halloysite nanotubes (HNTs), a natural clay mineral decorated with CuS, and a polydopamine (PDA) coating functionalised with the antimicrobial enzyme Lys. HNTs@CuS@PDA-Lys exhibited excellent bactericidal activity against *E. coli* (100.0 ± 0.2%) and *S. aureus* (99.9 ± 0.1%), eliminating 75.9 ± 2.0% of the *S. aureus* biofilm under NIR irradiation (808 nm, 1.5 W/cm^2^). Under NIR light exposure, a synthesised heterojunction composite of graphdiyne nanowall-wrapped hollow copper sulphide nanocubes (CuS@GDY) also presented strong localised surface plasmonic resonance and an enzyme mimic function [82]. The nanocomposite acts through the combined hyperthermic and increased peroxidase-like activities, facilitated by the exclusive hierarchical configuration, the tight bandgap of GDY nanowalls, the LSPR effect of CuS nanocages, fast interfacial electron transfer dynamics, and carbon Vs on CuS@GDY. Additionally, Cu_7_S_4_-2 with (224) facets showed outstanding antibacterial efficiency against *B. subtilis*, *E. coli*, and drug-resistant *P. aeruginosa* compared with Cu_7_S_4_-1 with (304) exposed facets via synergetic PDT and PTT [86]. 

Molybdenum disulphide (MoS_2_) belongs to the category of two-dimensional transition metal dichalcogenide nanosheets. Recently, its electronic structure was modulated by Vs engineering, with different concentrations of sulphur vacancies (Vs) being generated to optimise photothermal conversion efficiency. This strategy improves light absorption and prevents the recombination of photogenerated electron–hole pairs. MoS_2_ with abundant vacancies strongly binds to bacteria, inhibiting colony formation. Above a specific concentration, excessive Vs on the surface of MoS_2_ can be responsible for charge carrier blocking and a photothermal performance decrease [91,92]. The photothermal conversion efficiency (η) was 45.97%, and bacteria were eliminated under 808 nm NIR light irradiation. In another study, MoS_2_ nanosheets were doped with copper ions (MoS_2_@Cu^2+^) for reduced electron–hole recombination and improved photothermal efficiency [93].

A biodegradable multifunctional nickel sulphide (NiS_2_) nanozyme with photothermal performance, nano-catalysis properties, and glutathione (GSH)-depleting function was proposed in [94]. This nanomaterial showed very good photothermal performance, catalytic properties, good stability, and rapid metabolism, proving its peroxidase-like ability to kill bacteria. 

Biogenic copper selenide NPs (bio-CuSe) were incorporated into a polyvinylidene fluoride membrane to improve its qualities and antimicrobial properties [95]. NIR irradiation increased the water temperature near the membrane, allowing for >95% suppression of bacterial growth. The obtained conversion efficiency was 30.8%. 

Tin selenides (SnSe) with different morphologies (sphere, rod, plate, and surface wrinkled) were investigated as PTAs. Spherical SnSe showed the best antimicrobial performance through combined photothermal and photodynamic mechanisms and managed to eliminate 99.99% of *E. coli* and *B. subtilis* bacteria [96]. The best calculated photothermal conversion efficiency was 41.4%, higher than other published values. 

Ferrous-ferric oxide (Fe_3_O_4_) NPs have a strong enzyme-like catalytic ability in a wide pH range and can be used in a combined photothermal–enzymatic antibacterial treatment platform. The photothermal effect can increase the production of •OH from H_2_O_2_ through the Fenton reaction. Interestingly, the catalytic activity of Fe_3_O_4_ NPs intensifies with increasing temperature in the range of 25–50 °C [97]. In Vitro wound treatment with an NIR laser after adding H_2_O_2_ damaged the biofilm. Additionally, the combined treatments showed less wound inflammation after in vivo tests. Lv et al. synthesised magneto-plasmonic multi-branched Fe_3_O_4_@Au core@shell nanocomposites [98] with a photothermal conversion efficiency of 69.9%, complete bacterial ablation after NIR irradiation, good photostability, and several cycles of repeated use.

MXene materials, bidimensional transition metal carbides/nitrides, have gained increased attention since their discovery in 2011 [55]. MXenes exhibit hydrophilicity and outstanding photothermal conversion efficiency, which can lead to good antimicrobial activity [99]. Very recently, MXenes were tested for antimicrobial applications, and the results were unsatisfactory due to poor MXene–bacteria interactions and bacterial rebound in vivo. Ti_3_C_2_MXene was then used in a photothermal treatment and proved to have a unique membrane-disrupting effect, with sharp edges of nanosheets acting as “nanoknives” [100]. Different strategies were tried to improve antimicrobial efficiency. In [101], lysozyme was immobilised on titanium carbide Ti_3_C_2_TX MXene ultra-thin nanosheets modified with polydopamine for light-enhanced enzymatic inactivation of antibiotic-resistant bacteria due to the close contact between this antimicrobial material and bacteria. In another study, Ti_3_C_2_TX MXenes were combined with ciprofloxacin and incorporated into a hydrogel to trap and effectively kill all the tested bacteria [102]. An engineered interface between n-type Bi_2_S_3_ nanorods and Ti_3_C_2_Tx nanosheets produced more reactive oxygen species (ROS) due to the accelerated photogenerated charge separation and transfer due to the differences between their work function values. Bi_2_S_3_ NR grew directly on the surface of Ti_3_C_2_Tx nanosheets, leading to the generation of a potential contact difference and an increase in the local electron density on Ti_3_C_2_Tx, reducing the recombination of electron–hole pairs. This nanocomposite was a stable, biocompatible, highly effective antimicrobial with enhanced photocatalytic and photothermal properties [103]. 

#### 3.1.3. Other Inorganic PTAs 

Black phosphorus (BP) is a two-dimensional (2D) layered semiconductor material applied as a photothermal agent due to its high photothermal conversion efficiency, extinction coefficient, biocompatibility, and excellent biodegradability [104]. The photothermal performance of some BP-based PTAs are presented in Table 1. BP exhibits less cytotoxicity than graphene, but it is still more toxic compared to other 2D nanomaterials [105]. Even without NIR irradiation, BP nanosheets can cause physical damage to the bacterial membrane, RNA leakage, and death because of the sharp edges of the sheets [106]. As a disadvantage, BP nanosheets can undergo rapid oxidation and degradation in ambient environments [107]. The photothermal conversion efficiency of BP nanosheets can be further enhanced by conjugation with Au. This nanocomposite can destroy up to 58% of Enterococcus faecalis bacteria from a biofilm under NIR light irradiation [108]. BP nanosheets decorated with cationic carbon dots (CDs) acted against bacteria through photothermal treatment, photodynamic therapy, and electrostatic interactions between cationic CDs and bacterial walls [109]. CuS NPs were immobilised onto BP nanosheets, resulting in an efficient synergistic nanocomposite for fighting *P. aeruginosa* and *S. aureus* cells [110]. After a few minutes of NIR irradiation, the temperature rose by 30.4 °C due to the photothermal conversion efficiency of both CuS and BP nanosheets. Recently, Zhao et al. [111] fabricated antibacterial photothermal nanofibres composed of polycaprolactone (PCL), Ag NPs, and BP for infected wound healing. After irradiation, a significant increase in temperature up to 41 °C was registered, generated by BP, and this hyperthermia accelerates the movement of Ag+, preventing the formation of silver aggregates. In vivo studies indicated that the application of these complex nanofibres accelerated wound healing. Other studies proposed the conjugation of the two nanomaterials—BP and Ag NPs—in BP@AgNP nanohybrids with broadened visible light absorption [80] or a BP/Ag NP nanocomposite [112] with higher efficiency for Gram-positive bacteria than for Gram-negative bacteria. 

Amorphous red phosphorus (RP) has rarely been applied, despite its good biocompatibility. In one study, RP was used in a layered composite (with a graphene oxide layer on top), showing rapid and almost complete microbial inactivation under visible and NIR light [113].

Boron. A multifunctional nanoplatform was developed based on boron nanosheet (B NS)-coated quaternised chitosan (QCS) and the nitric oxide (NO) donor N,N′-di-sec-butyl-N,N′-dinitroso-1,4-phenylenediamine (BNN6). The B-QCS–BNN6 nanoplatform [114] exhibited photothermal therapy efficacy and provided controlled NO release after 808 nm laser irradiation, rapidly reaching >99.9% inactivation of bacteria.

**Table 1 ijms-24-09375-t001:** New representative inorganic-based PTA nanomaterials for antibacterial activity.

Type of Nanomaterial		Characterisation Morphology	Tested Bacteria	PTT Parameters	Performance	Ref.
Metal-based PTAs	Au NR	10 × 45 nm Au NRs attached to glass surfaces	*S. epidermidis* ATCC 35984	850 nm LED, I = 0.2 W·cm^−2^, 5 min	AR = 71% of biofilm Max—97%	[64]
	Au nanoworms covered with PDA	Nanoworms with diameters of 5 ± 1.5 nm, interconnected	*E. coli* *S. aureus*	808 nm I = 1 W·cm^−2^, 20 min 100 µg·mL^−1^ PTAs	∆T = 30.9 °C AR = 80% *E. coli* and AR = 90% *S. aureus*	[70]
	Glycomimetic polymers decorated Au NR	AuNR—50–100 nm long	drug-resistant *P. aeruginosa*	808 nm laser, I = 2 W·cm^−2^_,_ 5 min 125 μg·mL^−1^ PTAs	∆T = 15.4 °C AR = 80%	[74]
	Protease (bromelain)-conjugated AuNR	Au NR—32 nm length, 7.8 nm width	*E. coli* *S. aureus*	808 nm 50 μg·mL^−1^ PTAs	T_max_ = 66 °C AR = 96.8% *E. coli* AR = 97.9% *S. aureus*	[75]
	Peptide/neuropeptide-conjugated AuNR	Au NR—49 nm length and 11 nm width	*MRSA* *E. coli*	808 nm I = 2 W·cm^−2^, 4 min	T_max_~70 °C stable after 4 cycles AR = 99% for *MRSA* AR = 96% for *E. coli*	[76]
	AuAg yolk−shell cubic nanoframes	Well-defined cubic nanoframes 10 nm Au core and frame edge length: 25–60 nm; frame thickness: 3.8–6.1 nm Ag/Au ≈ 3:1	*MRSA* *E. faecalis* *P. aeruginosa* *K. pneumoniae* *B. bacillus* *E. coli*	808 nm laser I = 0.33 W·cm^−2^, 10 min	η = 65.6% at 0.27 W·cm^−2^; ΔT = 23.7 °C AR = 96.55%, *P. aeruginosa* AR = 93.69% *K. pneumoniae* AR = 92.34% *B. bacillus* AR = 96.73%, *E. coli* AR = 98.08% *E. faecalis*	[79]
	Fractal-like Ag nanoaggregates in SiO_2_ on poly (dimethylsiloxane) layer	AgNPs 10–20 nm, interNP distances of a few nm SiO_2_ = 1.3–25%	*S. aureus* and *E. coli*	808 nm laser I = 1.4 W·cm^−2^, 10 min; m = 15.4 μg PTAs	η = 50% AR = 100% of *S. aureus* biofilm (10 min) AR = 100% of *E. coli* biofilm (5 min.)	[77]
	Pd NPs	4 nm and 41 nm in diameter	*S. aureus* and *E. coli*.	808 nm laser, I = 1.35 W·cm^−2^, 10 min, 20 mg·L^−1^ PTAs	η = 33.1% AR = 99.99% *S. aureus* AR = 99.99% *E. coli.*	[84]
	Ag/Au bimetallic NPs on jellyfish nanofibre scaffold	Bimetallic Ag/AuNPs: nanospheres, nanotriangles	*B. subtilis* *P. aeruginosa* *E. coli*, *S. epidermidis*	808 nm laser, I = 1 W·cm^−2^, 5 min	T_max_ = 80 °C. Effective (AR = n.a.)	[86]
	Pd–Cu nanoalloy NPs+ AMO in ZIF-8	Spherical Pd–Cu nanoalloy NP size 9.02 nm	*S. aureus* *P. aeruginosa*	808 nm NIR laser, I = 1 W·cm^−2^, 10 min, 200 μg·mL^−1^ PTAs	η = 45.8% AR = 99.8% *S. aureus* AR = 99.1% *P.aeruginosa* CR = 75.3% *S. aureus* CR = 74.8% *P. aeruginosa*	[87]
Sulphides	Cu_7_S_4_ nanosheets	Cu_7_S_4_ samples with (224) exposed facets, a large number of nanosheets, diameter of 30–50 nm	*B. subtilis*, *E. coli* drug-resistant *P. aeruginosa*	808 nm laser, I = 1.5 W·cm^−2^, 10 min, 50 μg·mL^−1^ PTAs	η = 40.52% ∆T = 29.4 °C AR = 100% *E. coli* AR = 100% *B. Subtilis* AR > 90% *P. aeruginosa*	[91]
	CuS@GDY	Graphdiyne (GDY)-nanowall-wrapped hollow CuS nanocubes	*MRSA* and *E.coli*	808 nm laser, I = 0.4 W·cm^−2^, 10 min	η = 48%, ∆T = 28 °C AR > 99.999% *MRSA* AR > 99.999% *E.coli*	[82]
	CuS nanosheets with sulphur vacancies	Nanosheets: Diameter = 60–100 nm Thickness = 25–30 nm	*B. subtilis* and *E. coli*	808 nm laser, I = 1.2 W·cm^−2^, 10 min 50 μg·mL^−1^ PTAs	η = 41.8%, ∆T = 30 °C, AR = 99.999% (both)	[88]
	Sulphur-vacancy-modulated MoS_2_	Nanospheres—diameter 200–300 nm	*E. coli*.	808 nm laser, I = 1.5 W·cm^−2^; 10 min 50 μg·mL^−1^ PTAs	η = 45.97% ∆T = 32 °C ≈100% killed bacteria	[92]
	Cu-doped MoS_2_ nanoflowers	Nanospheres of 50–500 nm; Cu^2+^ uniformly distributed on the surface edge sites	*S. aureus*	660 nm laser, I = 0.898 W·cm^−2^, 20 min., 2 μg·mL^−1^ PTAs	∆T = 30.3 °C AR = 99.64%	[93]
	NiS_2_ nanozymes	Spherical NPs—diameter of 112 nm	*E. coli*, DH5α; *MRSA*, Mu50	808 nm laser, I = 0.75 W·cm^−2^, 10min, 75 μg·mL^−1^ PTAs	η = 43.8% ∆T = 23.4 °C AR = *E. coli* 98.33% AR≈92% *MRSA*	[94]
selenides	SnSe	Spherical particles	*E. coli* and *B. subtilis*	808 nm laser, I = 1.5 W·cm^−2^, 10 min 25 μg·mL^−1^ PTAs	η = 41.4% Tmax = 57 °C AR = 99.99% *E. coli* AR = 99.99% *B. subtilis*	[96]
	Cu_2_Se NPs in polyvinylidene fluoride membrane	80 nm size NPs	*E. coli* and *B. subtilis*	1064 nm laser, I = 2.0 W·cm^−2^, 400 s 160 μg·mL^−1^ PTAs	η = 30.8% ∆T = 14.6 °C AR = 97.52% *E. coli*	[95]
Oxides	Fe_3_O_4_ NPs	Mesoporous hollow Fe_3_O_4_ NPs	*E. coli* *S. aureus*	808 nm NIR + H_2_O_2_ (1mM) I = 1 W·cm^−2^, 10 min; 4 cycles 1 mg·mL^−1^ PTAs	η = 28.5% AR = 72% *S. aureus* and AR = 100% *E. coli*	[97]
	Magneto-plasmonic Fe_3_O_4_@Au core@shell	Fe_3_O_4_ spherical core and Au—branched structure	*E. coli* *S. aureus*	980 nm laser diode, I = 1.0 W·cm^−2^, 10 min, 50 μg·mL^−1^ PTAs	η = 69.9% AR = 100% *E. coli* and AR = 100% *S. aureus*	[98]
MXene	Ti_3_C_2_ MXene combined with ciprofloxacin	Ti_3_C_2_ nanosheet monolayer with 50–200 nm lateral size	*E. coli* *MRSA*	808 nm, I = 0.4 W·cm^−2^, 15 min, 100 μg·mL^−1^ Ti_3_C_2_+ 5 μg·mL^−1^ μg·mL^−1^ ciprofloxacin	Tmax = 60.7 °C AR = ≥99.99999%	[102]
	Ti_3_C_2_TX MXene-PDA-functionalised +lysozyme	Ti_3_C_2_ MXene—monolayer	*MRSA*	808 nm laser, I = 2.0 W·cm^−2^, 15 min. 50 μg·mL^−1^ PTAs	η = 46.88% Tmax = 63.5 °C. AR > 95% *MRSA*	[101]
	Bi_2_S_3_NR/Ti_3_C_2_Tx MXene	Ti_3_C_2_Tx Mxene few-layer nanosheets	*E. coli* *S. aureus*	808 nm light, I = 0.7 W·cm^−2^, 10 min	η = 35.43% T_max_ = 65 °C RA = 99.86% *S. aureus* RA = 99.92% *E. coli*	[103]
Other	BPs@cationic CDs	Few-layer or monolayer BPs with a flat structure, CDs (8–13 nm) grown in situ on BPs	*E. coli* *S. aureus*	660 nm + 808 nm lasers, I = 1.5 W·cm^−2^, 5 min, 200 μg·mL^−1^ PTAs	η = 34.1% ∆T = 28.2 °C RA ≈ 99% *S. aureus* and *E. coli*	[109]
	BPQDs@NH	BP quantum dots (BPQDs) of 3 nm encapsulated in hydrogel	*MRSA* AmpR *E. coli*	808 nm laser, I = 1 W·cm^−2^, 5 min, 200 μg·mL^−1^ PTAs	η = 42.9% ∆T = 35 °C RA = 90% *MRSA* RA = 90% AmpR *E. coli*	[115]

Abbreviations: I—laser power density (irradiance); AR—antibacterial rate; η—photothermal conversion efficiency; PDA—polydopamine; CR—clearance rate of biofilm; AmpR -ampicillin resistant; *S. epidermidis*—Staphylococcus epidermidis; *MRSA*—methicillin-resistant Staphylococcus aureus; *E. coli*—Escherichia coli; *B. subtilis*—Bacillus subtilis; *E. faecalis*—Enterococcus faecalis; *B. bacillus*—Bauman bacillus; *K. pneumoniae*—Klebsiella pneumoniae; *P. aeruginosa*—Pseudomonas aeruginosa; ZIF 8—zeolitic imidazolate framework-8; AMO—amoxicillin; CDs—carbon dots.

#### 3.1.4. Carbon-Based Nanomaterials

Carbon-based nanomaterials have attracted considerable attention as photothermal agents (PTAs) for antimicrobial applications owing to their distinctive structures and outstanding optical, thermal/electronic, and mechanical properties; versatility in functionalisation; high surface area [116]; deep tissue penetration; and reduced mammalian cytotoxicity [117]. Developing easy formulations of graphene-like materials could be an asset to the industry. For instance, highly reduced graphene oxide (HRG) via the reduction of graphite oxide (GO) is a versatile nanomaterial that can be functionalised as graphene-based inorganic nanocomposites for various applications [118]. 

Currently, from the entire carbon-based nanomaterial library, graphene-based nanomaterials (GBNs) and carbon nanotubes (CNTs) have become hot spots for eradicating and deactivating bacteria via various physical and chemical antibacterial mechanisms, such as chemical oxidation and ROS generation, the biological isolation of microbial cells, the generation of structural damage [52], and photothermal effects [119,120], mainly being investigated as antimicrobial PTAs. The antibacterial properties of GBNs appear to be influenced by the presence and number of functionalities on their surface [120]. At the same time, CNTs are characterised by size-dependent antibacterial activity that increases with a decrease in size [121]. Commercial green fluorescent graphene quantum dots (GQDs) were tested as a photoactive antimicrobial agent, and a heat yield of 50% (measured by the photothermal lens technique) was obtained under excitation at 532 nm (wavelength shorter than the emission band), proving their potential to be an efficient, safe, and low-cost photothermal agent [122]. Carbon dots have also been tested due to their biocompatibility and versatility. One study reports the utilisation of bacteria-affinitive carbon dots targeting the D-glutamic acid-adding enzyme (MurD ligase), which is involved in bacterial cell wall peptidoglycan synthesis [123]. Bacterial testing showed the ability of this material to kill 80.33% of *E. coli* and 89.27% of *S. aureus* without NIR light, and after only a few minutes of laser irradiation, more than 96% of *E. coli* and 100% of *S. aureus* were killed, proving the increased spatial accuracy of the antibacterial action with minimal cytotoxicity to human cell lines.

In fact, as PTAs for antimicrobial applications, carbon-based nanomaterials are usually combined with various compounds to improve the antibacterial performance through synergistic or additive effects, since their intrinsic photothermal properties, in some cases, are not sufficient to assure an appropriate antibacterial effect in a particular application. Thus, a great variety of reasonable carbon-based combinations with photothermal components (e.g., Au nanostars [124] and fluorophores [125]) and antibacterial compounds (e.g., Ag NPs [126,127]) have been designed and reported in the literature as efficient PTAs with adequate antibacterial activity in various practical applications. Several of the most representative C-based PTA nanomaterials for antibacterial activity are presented in Table 2. For instance, Oruc et al. [125] decorated the surface of multiwalled carbon nanotubes (MWNT) with NIR-absorbing 3,3′-diethylthiatricarbocyanine (DTTC) fluorophores to obtain efficient photothermal nanomaterials that can kill *Pseudomonas aeruginosa*. Under NIR irradiation, the formulated MWNT/DTTC nanohybrids could produce a powerful hyperthermal effect (the temperature of the dispersion reached around 92 °C after 15 min), leading to a 77% killing efficiency of *P. aeruginosa* cells. Then, the MWNT/DTTC nanohybrids were embedded within a waterborne polyurethane matrix. It was noted that under laser irradiation, the temperature increased to 120 °C, generating a substantial antibacterial and antibiofilm effect on *P. aeruginosa* cells attached to the surface. Further, Tan et al. [126] combined the excellent photothermal effect of RGO and intrinsic antibacterial features of AgNPs into an RGO/Ag nanocomposite to destroy both common bacteria (*E. coli*) and multidrug-resistant (MDR) bacteria (*K. pneumoniae*). Among the investigated samples, the RGO/Ag nanocomposite presented significantly higher antibacterial activity against both bacteria; this antibacterial activity synergistically increased under NIR irradiation (0.30 W/cm^2^ for 10 min) through the photothermal effect, which induced cell membrane disruption and the generation of ROS. In another work, Yang et al. [128] explored the synergy of the photocatalytic–photothermal effects of a stable BiOI-GO nanocomposite with better environmental disinfection properties, while Lv et al. [129] combined polyvinylpyrrolidone-functionalised AgNPs with reduced graphene oxide rGO (AgNPs-PVP@rGO) to fabricate a visible-light-triggered photoactive nanocomposite able to increase the visible-light-driven photocatalytic degradation and photothermal antibacterial activity.

Although carbon-based nanomaterials are characterised by a series of advantages, such as profuse sources, low cost, thermal and mechanical stability, good processability, and high thermal conductivity, besides biocompatibility issues, the relatively low photothermal effect for antibacterial activity represents their main drawback as PTAs, particularly when compared to the photothermal effects generated by inorganic-based or noble-metal-based nanomaterials.

In conclusion, plasmonic metals, especially AuNPs, distinguished among metal-based NPs and at the forefront of fighting bacteria, are easy to fabricate, stable, and biocompatible environment-responsive antimicrobial agents. In comparison, carbon-based nanomaterials, despite their versatility, have the disadvantages of biocompatibility issues and a relatively low photothermal effect for antibacterial activity.

### 3.2. Organic-Based PTAs

Recently, organic-compound-based nanomaterials have received increasing attention as potential alternatives to inorganic-based nanomaterials and have been extensively exploited in formulating PTAs with proper antibacterial activity. As PTAs, these compounds typically absorb photons produced by NIR irradiation and generate heat through nonradiative relaxation pathways. The category of organic-based PAT nanomaterials is generally represented by conjugated-polymer-based nanomaterials (e.g., polyaniline and polypyrrole), crystalline porous organic polymers (e.g., covalent organic framework), and polymer-functionalised nanomaterials. Some of the most recent and important research studies using organic-compound-based PTA nanomaterials for antibacterial activity are listed in Table 3.

#### 3.2.1. Conjugated Polymer (CP)-Based Nanomaterials

Among various classes of macromolecules, conjugated polymers (CPs) with absorption in the NIR range, such as polydopamine (PDA), polyaniline (PANI), polypyrrole (PPy), and poly(3,4-ethylenedioxythiophene) (PEDOT), are widely explored in designing new light-responsive nanomaterials with suitable antimicrobial and bactericidal performance. Besides the inherent electronic and optical features originating from the specific delocalised electronic structure and the presence of large π-conjugated backbones [121,132], CPs are characterised by low light scattering and a high penetration depth of NIR light into tissues, amenability in formulation, and higher biocompatibility than carbon-based nanomaterials, capable of mitigating their main drawback related to agglomeration [132]. To maximise the bacterial interaction capability, aqueous stability, and antimicrobial PT effect, CPs are usually modified with different compounds (e.g., cationic ammonium groups, PEI, Au nanorods, Au NPs, and magnetic NPs) (Table 3) [133]. Zhou et al. [134] formulated positively charged conjugated polymer (PTDBD)-based NPs with NIR-triggered activity and better bacterial interaction ability for antimicrobial therapy to advance phototherapy for bacterial infections. Under light with a low power light density of 1 W·cm^−2^ (808 nm) and a short time of 8 min, the simultaneous ROS and heat generated by the polymer PTDBD with a donor–acceptor (D–A) structure could effectively kill three representative microbes (i.e., *AmpR E. coli*, *S. aureus*, and *C. albicans*). Further, the authors investigated the efficacy of this strategy in vivo for treating *S. aureus*-infected wounds of mice and observed no significant damage to normal tissue, demonstrating its great potential in the application of treatments for bacterial infections. Later, Zhang et al. [87] used the same strategy to design cationic-conjugated PDTPBT NPs for photothermal antibacterial therapy under NIR light irradiation. Based on in vitro experiments, the constructed PDTPBT exhibited efficacious antibacterial ability upon 808 nm laser irradiation, in addition to excellent photostability and high photothermal conversion efficiency. In another work, Ko et al. [135] constructed a photothermal nanocomposite based on poly(3,4-ethylenedioxythiophene)–poly(styrene-sulphonate) (PEDOT: PSS) and agarose with thermo-processability, light-triggered self-healing, and excellent antibacterial activity. The authors demonstrated that during NIR exposure, PEDOT: PSS/agarose exhibited high shape flexibility through the NIR-light-induced self-healing effect after damage and excellent antibacterial activity against pathogenic bacteria, successfully destroying and killing *E. coli* and *S. aureus* within 2 min of irradiation.

#### 3.2.2. Polymer-Functionalised Nanomaterials

Functionalising nanomaterials with specific polymers (e.g., PEG, chitosan, or peptides) is a widely exploited strategy that, besides improving biocompatibility, dispersibility, protection in the biological environment, and specific targeting, may increase the physicochemical properties or endow the newly formulated nanomaterials with specific functions, advancing their effectiveness in PATs in practical antibacterial applications [136] (Table 3). In this respect, Fan et al. [137] managed to construct photothermal NPs that could efficiently kill *E. coli* at a relatively low temperature of ~45 °C under NIR irradiation by linking PDA NPs with thiolated poly-(ethylene glycol) (PEG) and magainin I (MagI) to increase the stability and bacterial interaction specificity. Jia et al. [138] constructed a versatile graphene-based photothermal nanocomposite that could rapidly and effectively eliminate Gram-positive *S. aureus* and Gram-negative *E. coli* bacteria, additionally destroying bacterial biofilms upon NIR irradiation. In this sense, the authors combined the efficient ability of chitosan to capture bacteria due to its positively charged functional groups with magnetic NPs and the photothermal conversion efficacy of GO. The formulated multifunctional nanocomposites could effectively eliminate bacteria after 10 min of NIR irradiation and destroy bacterial biofilms, suggesting their great potential in antibacterial applications. In another work aimed at resolving focal infections generated by antibiotic-resistant bacteria, Korupalli et al. [139] used the same strategy. They developed a pH-responsive polymer based on polyaniline-conjugated glycol chitosan (PANI-GCS) that undergoes self-assembly into NPs. The authors estimated that under NIR irradiation, the local temperature of PANI-GCS NPs increased by approximately 5 °C, leading to the specific and direct aggregation of bacteria, preventing tissue damage, and promoting wound healing. Furthermore, Wang et al. [140] proved that the functionalisation of photothermal-responsive conjugated polymer nanoparticles with cell-penetrating peptide (CPNs-Tat) might be considered a rapid and effective modality for combating bacterial infections. The positively charged Tat from the surface of NPs could efficiently enhance the interaction with bacterial cells, leading to CPNs-Tat/bacteria aggregation. At the same time, under NIR irradiation, CPNs-Tat could efficiently convert the light into heat and produce local hyperthermia to kill bacteria within a few minutes. 

#### 3.2.3. Covalent Organic Frameworks

Covalent organic frameworks (COFs) represent crystalline organic frameworks of porous polymers. Besides good thermal stability, reduced toxicity, and versatility in functionalisation, they contain specific light atoms (carbon, nitrogen, oxygen, and borane), with tailored and harmonious porosity [141,142]. These features enable them to be considered tremendous candidates for developing suitable platforms for application in different fields (e.g., gene and drug delivery, bioimaging, and biosensing) [141], mainly being used as wound-healing and antibacterial agents, owing to their long-lasting antibacterial properties and ability to interact with bacterial cells through their hydrophobic spatial structures [143]. In addition, the encompassed lightweight elements, strongly connected through covalent bonds, along with a specific 2D (two-dimensional) or 3D (three-dimensional) π-conjugation structure, make them critical light-activated agents for photothermal and photodynamic antibacterial effects (Table 3), as well as in combinatorial therapies [141,142,143,144]. A porphyrin-based COF (TP-Por-CON) containing a nitric oxide (NO) donor molecule, BNN6, within the pore volume of the framework structure for synergising photodynamic, photothermal, and gaseous therapies under red-light irradiation (635 nm) and efficiently killing Gram-negative bacteria (*E. coli*) and Gram-positive bacteria (*S. aureus*) in vitro was successfully synthesised [145]. Yang et al. [146] reported the construction of a covalent organic framework (TAPP-BDP) with a conjugated donor–acceptor structure. Under NIR irradiation (λ = 808 nm), it can exert triple and synergistic bacterial inhibition by combining photodynamic, photothermal, and peroxidase-like enzymatic activities. Based on in vitro investigations, the authors proved the excellent antibacterial efficiency of TAPP-BDP against Gram-negative and Gram-positive bacteria. At the same time, in vivo experiments further suggested the ability of the materials to heal wounds infected with *S. aureus* in animals. Recently, Li et al. [142] proposed a rational strategy for treating drug-resistant pathogenic bacterial infections by constructing a hydrogel with photocatalytic and anti-inflammatory activities based on a Cu co-coordinated D-A-type COF and sodium alginate hydrogel (CTCS) for the adequate healing of wound infections. Under NIR irradiation (λ = 660 nm), the CTCS hydrogel presented excellent bactericidal activity originating from the synergy of photothermal and photocatalytic effects, killing 99.95% and 98.5% of *S. aureus* and *E. coli* bacterial strains within the first 20 min. In vivo experiments confirmed that the CTCS hydrogel could be used as a strategy for the rapid reconstruction of bacterially infected tissues, owing to its ability to reduce the expression of TNF-α and promote wound healing and tissue regeneration (IL-10 and VEGF).

**Table 3 ijms-24-09375-t003:** Representative organic-based PTA nanomaterials for antibacterial activity.

	Matrix/ Material	Light (nm) and Intensity	Temperature Reached	Antibacterial Mechanism	In Vitro Biological Performance		Ref.
					Type of Bacteria	Efficacity	
CP	PTDBD	808 nm; 1.0 W/cm^2^	66 °C	PTT	*S. aureus* *E. coli* *C. albicans*	Effective	[134]
	PDTPBT	808 nm; 1.0 W/cm^2^	57 °C	PTT	*E. coli* *MRSA*	Effective	[87]
	PEDOT:PSS/agarose	808 nm; 2.0 W/cm^2^	24.5 °C	PTT	*S. aureus* *E. coli*	~100%	[135]
	PDPP3T	808 nm; 0.50 W/cm^2^	~45 °C	PTT	*E. coli*	~100%	[147]
	DMCPNs	808 nm; 0.50 W/cm^2^	62.4 °C	PTT and PDT	*E. coli*	93%	[148]
Polymer functionalised nanomaterials	MagI-PEG@PDA NPs	808 nm; 2.0 W/cm^2^	45 °C	PTT	*E. coli*	99.99%	[137]
	GO–IO–CS nanocomposite	808 nm; 2.0 W/cm^2^	~25 °C	PTT and capture bacteria and aggregation	*S. aureus* *E. coli*	~80%	[138]
	CPNs-Tat	808 nm; 2.0 W/cm^2^	55.3 °C	PTT	*E. coli* *S. aureus* *C. albicans*	~100%	[140]
	SF-CS-PDA cryogels	808 nm; 2.0 W/cm^2^	~45 °C	PTT and ROS-scavenging capacity, tissue affinity	*S. aureus* *E. coli*	Effective	[149]
COFs	TP-Por-CON@BNN6	635 nm	-	PTT and PDT and gaseous therapy	*S. aureus* *E. coli*	Effective	[145]
	TAPP-BDP	808 nm	65 °C	PTT and PDT and ROS	*S. aureus* *E. coli*	Effective	[146]
	CTCS	660 nm 0.4 W/cm^2^	~54 °C	PTT and PDT	*S. aureus* *E. coli*	>98.5%	[142]

Abbreviations: PTDBD—positively charged conjugated polymer; PDTPBT—cationic water-soluble conjugated polymer based on a donor–acceptor (D–A) structure; PEDOT:PSS/agarose—poly(3,4-ethylenedioxythiophene)–poly (styrene-sulphonate)/agarose nanocomposite; PDPP3T—diketopyrrolopyrrole-based conjugated polymer; DMCPNs—dual-mode conjugated polymer nanoparticles based on poly(diketopyrrolopyrrole-thienothiophene) (PDPPTT) and poly[2-methoxy5-((2-ethylhexyl)oxy)-p-phenylenevinylene] (MEH-PPV); MagI-PEG@PDA NOPs—magainin-modified polydopamine nanoparticles; GO–IO–CS—chitosan-iron oxide—functionalised magnetic graphene oxide; CPNs-Tat—photothermal-responsive conjugated polymer nanoparticles functionalised with cell-penetrating peptide; SF-CS-PDA—polydopamine nanoparticles incorporated into chitosan/silk fibroin cryogel; TP-Por-CON@BNN6—porphyrin-based covalent organic framework containing nitric oxide and BNN6; TAPP-BDP—covalent organic framework with a conjugated donor–acceptor (D–A) structure; CTCS—Cu co-coordinated D-A-type COF and sodium alginate hydrogel.

Although the photothermal effect of organic-based nanomaterials usually does not outperform that of inorganic materials, these materials have attracted tremendous attention as PTAs, owing to their biocompatibility and potential biodegradability, essential features that are missing in the case of inorganic materials and that can be further fine-tuned depending on the targeted application. 

Among investigated materials, CP and COFs are characterised by relatively good biocompatibility, significant absorption coefficients, and high photothermal conversion efficiency. In contrast, besides their acceptable biocompatibility, functionalised polymer nanomaterials are endowed with specific targeting segments, which may resolve the most frequently faced challenge of nanomaterials and the agglomeration process and increase the PTT performance as an antibacterial. 

Despite many promising outcomes of organic-based PTT, there are still practical barriers to clinical translation. First, their synthesis/formulation can be expensive and laborious, so simple preparation methods for scale-up are still needed. Second, in vitro and in vivo studies related to long-term biosafety are still in their infancy and are challenging. At the same time, the biodegradation mechanism of complex organic structures, such as PTAs, in living organisms still needs to be investigated. Therefore, further investigation is required to design more biocompatible organic-based PATs with predictable biodegradation mechanisms and biological behaviour that would satisfy PTT efficacy.

In conclusion, among organic PTAs, covalent organic frameworks (COFs), due to their good thermal stability, reduced toxicity, and versatility in functionalisation, are considered tremendous candidates for developing suitable platforms for application in different fields.

### 3.3. Hybrid Photothermal Antimicrobials and Inorganic–Organic Nanocomposites

Metal–organic framework (MOF)-derived hybrid materials have been developed as promising multifunctional nanomaterials or nanocarriers for medical applications such as diagnosis and antimicrobial therapy [150]. Moreover, NPs can be incorporated into hydrogels and used as nanocomposite hydrogels. NPs can be added directly to hydrogels, produced in situ via a reaction within the hydrogels, or mixed with a hydrogel precursor to undergo gelation and form the final NP hydrogel. Interestingly, nanocomposites’ high chemical or physical complexity allows synergistic effects and better functionality [151]. Therefore, hybrid nanosystems have been increasingly developed for their versatility and efficacy in overcoming obstacles not readily surmounted by their nonhybridised counterparts. 

For instance, rough-surface nanoparticles with satisfactory biocompatibility, such as carbon–iron oxide nanohybrids with rough surfaces (RCF) [152] or NiFe_2_O_4_@Au/PDA [153], demonstrated antibacterial effects via synergistic photothermal therapy (PTT)/chemodynamic therapy (CDT) effects in the NIR-II bio-window and photothermal–magnetolytic effects, respectively. The nanostructures presented increased bacterial adhesion for effective interaction, a better penetration depth, and a low power density in in vitro and in vivo studies against *E. coli*, *S. Aureus*, and *MRSA*. Excellent antibacterial activity against S. aureus (99.7%) and *P. aeruginosa* (99.9%) occurred with the heat-induced release of the antimicrobial agent physcion (Phy) from drug-loaded black phosphorus nanosheets (BPNSs@phy) [154]. The BPNSs presented excellent photothermal conversion ability, which disturbed the hydrophobic interactions that kept the antibiotic on the nanosheets and facilitated Phy release, thus demonstrating PTT/CDT synergism for a better bactericidal effect.

Furthermore, loading hydrogels with nanoparticles increased their functionality. Fluorescent carbon dots (CDs) employed as carriers for curcumin (Cur) within a CD/Cur nanocomposite [155] exhibited low cytotoxicity and negligible haemolytic activity. IK8-liposome/AuNR-loaded hydrogels [156] were fabricated by incorporating antimicrobial-peptide-loaded liposomes, IRIKIRIK-CONH_2_ (IK8), and gold nanorods (AuNRs) into poly(ethylene glycol) (PEG) to protect them from proteolysis and to employ the PTT capacity for a controllable PTT/CDT synergistically enhanced antibacterial nanoplatform against S. aureus and Pseudomonas aeruginosa. 

Enhancing the antibacterial activity of silver ions (Ag^+^) was possible through a silver-nanoparticle-embedded carrageenan hydrogel, gallic acid-modified silver nanoparticles (GA-Ag NPs Carr) [93], and antimicrobial-peptide–gold/silver nanorods (Dap@Au/Ag NRs) [157], which destroyed the integrity of the *MRSA* membrane, resulting in content leakage and bacterial death. The platforms exhibited PTT/CDT-enhanced antibacterial activity via Ag^+^ released from the NPs and NRs and the NIR-laser-induced photothermal assistance of GA-Ag NPs and Au/Ag NRs. The hydrogels also presented good biocompatibility and effective anti-*S. aureus*, anti-*MRSA*, and anti-*E. coli* activity and healing-promoting properties in vivo. Similarly, wound healing was accelerated in diabetic rats when studying a black phosphorus quantum-dot-based hydrogel (BPQDs@NH) [158]. *MRSA*-infected wounds exposed to combined PDT/PTT were effectively sterilised due to the rapid increase in temperature (up to 55 °C), ROS production, lipid peroxidation, glutathione, adenosine triphosphate accumulation, and bacterial membrane destruction. The enhanced photocatalytic and photothermal performance of molybdenum disulphide (MoS_2_) nanosheets doped with copper ions (MoS_2_@Cu^2+^) resulted in 99.64% efficacy against *S. aureus* [93]. The underlying MOA consists of a combination of hyperthermia, ROS, and Cu^2+^ release. Cu^2+^, by absorbing photons and converting the photoenergy into heat (the d–d transition of electrons), contributes to intense PTT. At the same time, Cu^2+^ also absorbs photogenerated electrons from MoS_2_ and contributes to enhanced ROS (reducing electron–hole recombination). Despite the promising initial results, developing effective MoS_2_-based antibacterial nanomaterials is still problematic due to the hydrophobicity and the weak interaction with bacteria and ROS. Therefore, constructing a polyethylenimine-modified molybdenum disulphide (MoS_2_-PEI) nanocomposite enhanced the stability and promoted the binding to the surface of bacteria through electrostatic interactions for enhanced photothermal antibacterial activity [159] and even enabled combined chemo/photothermal/photodynamic triple-mode therapy for bacterial and biofilm infections [160]. Under NIR light irradiation, MoS_2_-PEI exhibited evident synergistic antibacterial efficacy against *E. coli* and *S. aureus* with a long-term bactericidal effect. High-efficiency bactericidal and long-term bacteriostatic effects with less bacterial rebound were observed in an *MRSA*-induced murine abscess under PTT with a Ti_3_C_2_MXene-based hybrid hydrogel. The rationally designed MXene-based hybrid hydrogels provided a strategy for cost-effectively treating localised bacterial infections using nanosystems [102]. Exploiting the photothermal sensitivity and peroxidase-like activity against one strain of a vancomycin-intermediate *S. aureus* reference strain and *E. coli* proved successful due to the encapsulated tungsten sulphide quantum dots (WS_2_QDs) and vancomycin (VAN) in thermal-sensitive liposomes. Interestingly, the enzymatic properties of WS_2_QDs, both the intrinsic and temperature-dependent ones, contributed to the improved CDT efficacy, illustrating the platform’s potential as one controllable system. The nanosystem also had antibiofilm properties via biofilm disruption for better drug transmembrane passage. Moreover, the in vivo studies highlighted its biocompatibility and the possibility of exerting the synergistic chemodynamic/photothermal antibacterial effects as reliable therapeutic approaches [161]. One complex nanoplatform based on a hybrid structure was proposed as a novel therapeutic option for *MRSA* skin infections. In this case, the system incorporated two-layered microneedle (MN) arrays: one water-insoluble inner layer with NIR photothermal capacity was encased by one water-soluble external layer loaded with vancomycin (VAN). The photothermal core comprised flame-made plasmonic Au/SiO_2_ nanoaggregates and polymethylmethacrylate (PMMA). The evaluation showed a synergistic CDT/PTT (VAN and heat above 55 °C for 10 min) effect, which reduced methicillin-resistant *S. aureus* (*MRSA*) survival by up to 80% [162]. The antibacterial and wound-healing capacities of injectable and self-healing hybrid hydrogels showed high-efficiency photothermal antisepsis under mild PTT conditions. A hybrid hydrogel prepared by the self-polymerisation of dopamine into polydopamine and the synchronised reduction of Ag^+^ to Ag NPs within a chitosan scaffold presented spontaneous recovery after mechanical damage, maintained structural integrity, and recovered the original admirable antimicrobial functions in vitro and in vivo with no obvious toxicity [163]. Since the toxicity of certain nanocomposites, such as AuNPs, needs to be mitigated prior to incorporating them into nanoplatforms for biological use, one strategy was proposed: having the AuNPs immobilised onto a larger particulate system, natural clay halloysite nanotubes (HNTs), with the HNTs modified with antibodies against *Escherichia coli* (*E. coli*, as a model microorganism) for immune-targeted PTT. The resulting AuNR-Ab-HNT hybrids demonstrated that harnessing antibody-functionalised HNTs as carriers increases the potential of the functionalised PTT/immunotherapy nanoplatforms for the targeted delivery of antibacterial nanoparticle combinations (e.g., silver or metal oxides) or antibiotics for localised antimicrobial infections [164]. 

In conclusion, advances in nano-biotechnology are promising and pave the way towards NIR-controlled multimodal potent antibacterial hybrid platforms without apparent toxicity. Designing and manufacturing intelligent nanosystems as more effective and selective alternatives will address the worldwide expansion of antibiotic-resistant species and the need to protect the microflora from non-specific antibiotics.

## 4. Applications

There are many expectations that PTAs must meet to comply with the essential criteria for future clinical implementation. There are vital technological and pharmacological requirements, from finding suitable nanomaterials and incorporating them into easy-to-manufacture devices to efficacy, efficiency, and biosafety to cost-effectiveness and user-friendliness. In the case of potential applications, the examples that follow also highlight the need for standardisation, despite the difficulties in managing the wide variety of nanomaterials used in various conditions, including the concentration [124], power density, wavelength, laser light’s power, exposure time, focal spot size [72,73], type of bacteria, and antibacterial MOA for the best efficacy and highest safety to the tissues [6,8,11,18,19,20]. The ideal will be intelligent nanosystems to respond to the microenvironment and deliver safely. Figure 2 depicts the roadmap of PTA-based nanoplatforms towards clinical implementation. The ideal antimicrobial platform will be multimodal, highly efficacious, biocompatible, easy to fabricate, included in standardised protocols, and cost-effective.

### 4.1. Antibacterial Biofilms

Biofilms are structured 3D complexes of grouped bacteria adherent to a surface and embedded in an autogenerated matrix of extracellular polymeric substances. The biofilm matrix comprises various extracellular polymeric substances (EPSs), such as polysaccharides, proteins, amyloids, lipids, extracellular DNA (eDNA), membrane vesicles, and humic-like microbially derived refractory substances [165]. Biofilms enhance antimicrobial resistance via mechanisms yet to be elucidated [166]; thus, nanoparticles’ antibiofilm capacity depends on many factors, such as biofilm maturity, surface composition and chemistry, nanoparticle size, surface charge, surface chemistry, and nanoparticle concentration [167]. Generally, antibiofilm nanomaterials either destroy the biofilm or interfere with biofilm formation. Interfering with biofilm formation [158] could be one method to address bacterial infections and wound healing [168]. Moreover, penetrating the almost 50 µm thick infectious biofilms [21] requires transporting the antimicrobials through hydrophilic biofilm channels [22,23,24] without adsorbing to the channels’ walls [25]. In addition, nano-antimicrobials should resist reticuloendothelial rejection during transport within the vascular system [26,27], which limits their operative size to between 100 and 200 nm [28]. Therefore, the ideal PTT-based antibacterial and anti-inflammatory photothermal agents (PTAs) should present high photothermal conversion efficiency and stability, good biosafety, responsiveness to the microenvironment [169], and cost-efficient fabrication [170] via the right combinations of antimicrobial mechanisms [171,172] and patient-friendly devices [173]. 

In Vitro and in vivo experiments specified the success rates and challenges of PTAs as antibiofilm agents. BSA@MPN + NIR treatment induced long-term varying degrees of bacterial membrane malformations and achieved >99% eradication of biofilms of *S. aureus* and *E. coli* [174]. Meanwhile, graphene and its derivatives, graphene oxide (GO) and reduced GO (rGO), due to their intrinsic properties and functionalisation with metal NPs, natural compounds, and antibiotics, could damage the bacterial morphology, release intracellular substances, and destroy the biofilm. The agglomerated structure of GO hydrogels (i.e., chitosan, collagen, or polyvinyl alcohol) could entrap and stack bacteria, preventing their initial attachment and biofilm formation. The sharp edges of GO could destroy the extracellular polymeric substance surrounding the biofilm and ruin the biofilm biomass structure [175]. Ag+ released from Ag NPs in Ag-NP-incorporated quaternised chitin (DQCA) nanomicelles [81] interacts with proteins and enzymes and significantly deforms bacterial membrane structures. At the same time, the high concentrations of ROS produced perturb cellular metabolism [176,177]. Ag^+^-GCS-PDA@GNRs [178] rapidly releases Ag^+^ in a pH-controlled manner to increase the bacterial membrane permeability, pierce them, even at a very low dosage, and thermally damage the membranes of Gram (+) and Gram (-) bacteria. Interestingly, local hyperthermia increases the Ag^+^ release concurrently and further improves the nanoplatform’s chemotherapeutic effect via a synergistic antibacterial mechanism. Unfortunately, despite the superior antibacterial properties of Ag NPs, the high cost and toxicity to humans (e.g., argyria, muscle spasms, and gastrointestinal disorders) have limited larger in vivo applications [179,180]. Upon NIR irradiation, nanocomposites with a gold core and copper (I, II) sulphide shell (Au@Cu_2–X_S) were shown to destroy *Enterococcus faecalis* and *Fusobacterium nucleus* biofilms through the decomposition of microbial exopolysaccharides during photothermal and peroxidase-like catalytic activity [181]. Interestingly, a low exogenous NO concentration can enter bacteria and provide a degree of antimicrobial activity through physical and functional changes [182]. NO activation and less intense local hyperthermia (<45 °C) represented the primary mechanism of effective biofilm elimination in vivo. Importantly, AI-MPDA acted as an all-in-one cytocompatible platform via NO-enhanced PDT, while low-temperature PTT severely disrupted the bacterial membranes and prevented bacterial colonisation [183]. In Cip-Ti_3_C_2_ TSG, “nano knives” and PTT led to membrane damage and could improve the penetration of Cip to achieve high-efficiency sterilisation. In addition, the functionalised Ti_3_C_2_ nanocomposites with cationic Cip can combine with the bacterial membrane through electrostatic interactions, which was conducive to the capture and killing of *MRSA* [102]. Among the new strategies, using dissolvable microneedle (MN) patches was proven to have potential. α-Amylase-PDA@Levo microneedles, fabricated via a two-casting method, incorporated levofloxacin dopamine NPs (PDA@levo) and α-amylase as the active ingredients and polyvinyl alcohol (PVA) as the fast-dissolving matrix. Under NIR, the MNs effectively delivered the enzymes, antibiotics, and PTAs into the cellular membranes. Enzymolysis destroyed the structure of the EPS matrix (extracellular polysaccharides) to eradicate biofilms, while PDA@Levo nanoparticles eradicated biofilms and killed the exposed bacteria via synergistic chemotherapy–PTT. The entire process also reduced the inflammation time and promoted wound healing and tissue regeneration [184].

As mentioned previously, interfering with bacterial physiological functions is also one possible MOA of the NIR-activatable antibiofilm activity of PTAs. One study revealed how deoxyribonuclease (DNase)-carbon monoxide (CO)@mesoporous polydopamine nanoparticles (MPDA NPs) efficiently eliminated *MRSA* biofilms through DNA degradation, and microbial destruction was caused by CO gas molecules [185]. A titanium implant was covered with black phosphorus and a complex hydrogel formed by poly(vinyl alcohol) modified with chitosan, polydopamine, and a nitric oxide-releasing donor to eradicate *MRSA* biofilm and to support osteogenesis. NIR light irradiation generated peroxynitrite (•ONOO^–^), which impacted the gene regulation of biofilm formation factors (intercellular adhesion gene D-icaD; intercellular adhesion gene A-icaA; staphylococcal accessory regulator -SarA), as well as virulence factors (α-haemolysis, staphylococcal enterotoxin A), halting *MRSA* biofilm formation [186]. A protease-conjugated AuNR antibacterial system reduced surviving bacterial populations to 3.2% and 2.1% of untreated control numbers for *E. coli* and *S. aureus*, respectively, and inhibited biofilm formation and exotoxin secretion even in the absence of NIR radiation. However, the enhanced degradation of existing biofilm and exotoxin was observed when PGs were used with NIR laser irradiation. This promising new strategy achieved both a reduction in viable microorganisms and the elimination of biofilm and exotoxin. Thus, this strategy addresses the long-ignored issue of the persistence of bacterial residues that perpetuate chronic illness in patients even after viable bacteria have been eradicated [75]. Intriguingly, the enhanced protease stability due to immobilisation may protect the enzyme from inactivation and would boost the enzymatic degradation of bacterial surface transmembrane proteins or signal molecules (such as autoinducer peptide—AIP) to further reduce bacterial viability, even at suboptimal temperatures. Moreover, according to previous studies, the photothermal effect, regarded as an internal heating model, could boost the activity of conjugated enzymes, which may lead to a synergistic effect [187].

Notably, functional coatings using immobilised photothermal agents are efficient means of sterilisation by breaking down and stopping biofilm formation. These compounds target quorum-sensing (QS) molecules and virulence factors and disturb the essential intercellular signalling mechanism, which regulates biofilm formation, virulence, the formation of spores or fruiting bodies, apoptosis, and genetic competence [188]. Furthermore, the combined modalities were proven to have effective antibacterial potential through synergistic PTT, PDT, and chemotherapy effects for inhibiting biofilm formation and killing deep biofilm bacterial cells [160]. 

### 4.2. Synergistic Photothermal-Effect-Based Antibacterial Systems

The effects of PTT-induced hyperthermia on the healthy surrounding tissues [189,190], hypoxia in the deep infection microenvironment, which reduces PTT efficiency [191,192], the excess ROS causing inflammation, fibrosis, and necrosis of normal cells [193], and the low catalytic activity of CDT [194,195] are limiting factors for its therapeutic effects in vivo. Since combination therapy is widely adopted in bacterial treatment, developing synergistic modalities for bacterial elimination has enormous prospects in biomedical applications. The advantages of the two combined treatment methods complement and reinforce each other, leading to an effect of “1 + 1 > 2”. Consequently, integrating multiple antibacterial mechanisms shortens the time to antibacterial onset, improves the antibacterial efficiency, and reduces the dose of antibacterial agents. Synergistic photothermal antimicrobial therapy primarily involves photodynamic–photothermal therapy, chemo–photothermal therapy, and nitric oxide (NO)–photothermal therapy. 

#### 4.2.1. Antibacterial Photothermal Therapy (PTT)–Photodynamic Therapy (PDT) 

Photothermal–photodynamic antibacterial therapy combines PTT and photodynamic therapy (PDT) to kill bacteria with high temperatures and reactive oxygen. Antibacterial PTT-PDT synergism reduces cell activity through PTT-induced local hyperthermia, increases cell sensitivity to the ROS generated through PDT, and inactivates the cells [196]. Several studies proposed synergistic models following the principles of cost-efficiency and biocompatibility. Quaternised chitosan (QCS)/Ag/CoP nanoplatforms demonstrated fast and efficient antibacterial properties while being nontoxic to mammals. Moreover, the QCS/Ag/CoP nanocomposites inactivated more than 99.6% of *S. aureus* and *E. coli* at very low concentrations (50 μg/mL) within 10–15 min due to the synergistic effects of the components. Ag enhanced the photocatalytic and photothermal effects of CoP, and the QCS coating improved the water dispersibility to provide better contact between the antiseptics and bacteria [197]. Carbon-based materials that have good biocompatibility and are environmentally friendly were considered antibacterial agents for synergism. For instance, modifying the surface of GO with CuS NPs improved PDT efficiency under NIR laser irradiation.

Furthermore, the antibacterial activities of GO and GO@CuS were less than the tri-modal synergistic GO-Tobramycin (Tob) and GO-Tob@CuS nanoplatforms, with excellent photothermal conversion capabilities and efficiency against antibiotic-resistant Pseudomonas aeruginosa (*P. aeruginosa*) and *S. aureus* models [198]. However, the registered toxic effects limited their further implementation [199]. Benefiting from their excellent absorption of NIR, carbon dots (CDs) exhibit a competitive NIR-laser-induced photothermal effect that enables the direct killing of bacteria through hyperthermia from PTT [116]. Since the CDs’ antibacterial activity was considered insufficient, the combination of copper ions with NIR-emitting CDs (RCDs) (as Cu-RCDs) and quaternary amino compounds (QACs) (as Cu-RCDs-C_35_) achieved a better antibacterial effect against Gram-positive and -negative bacteria as tri-modal (photothermal, photodynamic, and quaternary ammonium salts) synergistic platforms [200]. 

To potentiate noble metals, such as gold, silver, and palladium, or polyphenolic substances, such as curcumin (Cur), which are usually considered photothermal antibacterial agents, loading them on small molecules or macromolecule carriers to form nanoagents was considered. For instance, mesoporous-silica-modified AuNRs, used as carriers, were loaded with Cur to construct a multifunctional composite antibacterial nanosystem (AuNRs@Cur), which had significantly improved PTT-PDT antibacterial effects compared to each incorporated photosensitiser and had insignificant cytotoxicity and haemolytic activity [201]. However, noble metals are imperfect due to their intrinsic features, such as poor photostability and biocompatibility, complicated preparation, high cost, and low antibacterial efficiency in vivo. Therefore, an alternative, MoS_2_/ICG/Ag^+^NIR, was proposed as another tri-modal synergistic combination PTT/PDT/chemotherapy treatment group with better in vivo results: the survival rates of both *S. aureus* and *E. coli* were close to zero, indicating that MoS_2_/ICG/Ag had the best broad-spectrum antibacterial activity under NIR light irradiation at 808 nm [160]. In the case of combined MoS_2_ and TiO_2_ within a transition metal sulphide/TiO_2_ nanofibre platform (MoS_2_/TiO_2_ NFs), the photothermal effect of the 3D/2D heterostructure (MoS_2_/TiO_2_ NFs) significantly improved, with a rapid increase in the local temperature to above 50 °C to inactivate bacterial proteins. The co-irradiation and oxidase-like synergistic antibacterial platform also increased the permeability of bacterial cell membranes via PTT to increase the membrane permeability for VIS/NIR-activated ROS and led to bacterial oxidative stress, the significant leakage of bacterial contents, the peroxidation of bacterial antioxidants, and eventually the death of bacteria. The platform effectively promoted *S. aureus*-infected wound healing with negligible haemolytic activity and cytotoxicity in mammalian cell lines [202]. Meanwhile, other studies used copper sulphide nanoparticles (CuSNPs) as a new class of low-cost PTT and PDT materials with a strong local thermal effect and a large amount of ROS under NIR irradiation that could cause bacterial oxidative damage [203]. Enhancing the antioxidant effect in vitro and in vivo via synergistic local PTT-PDT was possible with AgNPs as PAM–PDA/Ag@AgCl [204] and (AgNPs@TA) hydrogels [205]. Furthermore, some antibacterial treatment platforms, known as photothermal nanozymes, combine nanoenzymes with peroxidase-like catalytic activity and photothermal effects for antibacterial therapy. IONPs employing synthesised iron oxide (Fe_3_O_4_) nanoparticles have good biosafety, excellent photothermal conversion ability, and peroxidase-like catalytic activity. The production of •OH in a slightly acidic environment achieves specific bactericidal effects and increases the sensitivity of bacteria to heat, thus synergising with PTT. 

Interestingly, the reactions stimulated one another, as demonstrated by the excellent antibacterial rate of *E. coli* and *S. aureus* in vitro. An in vivo study on *S. aureus*-infected wounds of mice demonstrated that IONPs effectively promoted healing and had clinical potential as anti-infection therapy [97]. Anti-infection therapy based on enhanced photocatalytic bactericidal activity could be achieved with nanocomposite hydrogels, which showed excellent photothermal properties. The resultant effect was attributed to the combination of polydopamine (PDA) and the natural antioxidant tannic acid (TA), as demonstrated in vivo on an *S. aureus* and *E. coli* co-infected skin wound model. A nanocomposite hydrogel incorporated polydopamine (PDA) for biocompatibility and adhesion. In the case of nanocatalysed hydrogels with an activated infection microenvironment response, using polyvinyl alcohol as a scaffold and MXene/CuS bio-heterojunctions for synergistic PTT-PDT effects, hyperthermia and NIR-light-generated singlet oxygen and hydroxyl radicals induced good antioxidant and antibacterial properties. This approach supported the enhancement of phototherapeutic effects in wound infection treatment [206]. Remarkably, infections are caused by aerobic and anaerobic bacteria, and research efforts are focused on related solutions. For instance, in a hypoxic environment in vivo, porphyrin from designed macromolecular compounds (e.g., TMPyP, TMPyP/(CB[7])_4_ [207], TP-Por CON, TP-Por CON@BNN6 [145]) could be reduced to phlorin by some facultative anaerobic bacteria with a strong reduction ability, such as *E. coli* and *S. typhi*, and acts via PTT as a good antibacterial. However, in an aerobic environment, where aerobic bacteria such as *B. subtilis* and *P. aeruginosa* were not reduced, TMPyP was a typical photosensitiser that could effectively kill bacteria through PDT. Therefore, in one environment, the best porphyrin compound may simultaneously play a role in the synergistic PTT–PDT effect to improve the antibacterial effect and reduce the side effects of a single treatment while having good biocompatibility. Moreover, AgNPs combined with GO and exposed to NIR irradiation exhibited increased photothermal activity, generating ROS and disrupting the microbial membrane in *E. coli* and *K. pneumoniae* [126]. A simple one-pot hydrothermal process successfully synthesised a flower-like CuS/GO hybrid. GO served as a perfect electron acceptor, transported the photogenerated electrons from CuS, and efficiently suppressed the recombination of hole–electron pairs, thus enhancing the photocatalytic properties. Moreover, the CuS and GO structural characteristics also improved the hybrid’s photocatalytic functioning. Consequently, the synergistic photothermal–photocatalytic–Cu^2+^-releasing effects of the CuS/GO-based nanosystem contributed to significant antibacterial efficacy under NIR irradiation for 15 min. Furthermore, the hybrid presented pronounced biocompatibility [208].

#### 4.2.2. Antibacterial Photothermal Therapy (PTT)–Nitric Oxide (NO)-Releasing Therapy

NO-releasing–photothermal antibacterial therapy combines photothermal agents (PTAs) with NO donor materials for higher bactericidal efficiency [183]. NO has been recognised as a broad-spectrum antibacterial agent with various MOAs, such as inducing lipid peroxidation to damage bacterial membranes, enhancing RNS production to perturb bacterial metabolism, or triggering severe oxidative stress for DNA cleavage [209]. Zhao et al. [210] combined SNOs with thiolated graphene (TG) and 4-mercaptophenyl boronic acid and added the composite to the surface of TG-NO to obtain one new biocompatible combination, TG-NO-B. The boric acid groups in TG-NO-B are covalently linked to the bacterial lipopolysaccharide units of bacterial cells and their biofilm matrix. They conferred reasonable specificity for Gram-negative bacteria in vivo and in vitro. Moreover, intermittent laser irradiation (30 s every 5 min) produced a NO-controlled release mechanism. Therefore, TG-NO-B significantly improved the antibacterial efficiency and reduced adverse side effects on surrounding healthy tissues. Additionally, controllable NO release was observed with Fe_3_O_4_@PDA@PAMAM@NONOate under intermittent 808 nm laser irradiation. Fe_3_O_4_@PDA@PAMAM-G3 expressed a concentration-dependent photothermal effect and high photothermal stability with accelerated NO release under NIR through PTT for anti-*E. coli* and anti-*S. aureus* effects. However, the difference in NO-releasing activity may be caused by the additional outer membrane barrier of Gram-negative bacilli, making them less sensitive to NO. One more practical approach is the excellent magnetic properties of Fe_3_O_4_@PDA@PAMAM@NONOates, which may be a way to rapidly remove bacteria in vitro with an external magnet [211]. The controllable synergistic PTT/NO activity of the MoS_2_-BNN6 platform resulted in timely and efficient antibacterial effects against ampicillin-resistant *E. coli*, heat-resistant *Escherichia faecalis* (*E. faecalis*), and *S. aureus*. Notably, the platform worked to selectively enhance oxidative/nitrosative stress and even DNA damage, accelerate glutathione oxidation, and subsequently reduce the usage of ROS/RNS generated in bacteria [212]. Being effective even at low concentrations proved to be one crucial asset. In the case of GNS/HPDA-BNN6, synergistic PTT-NO effectively destroyed bacterial biofilms even at concentrations less than 200 mg/mL. The gold nanostar/hollow dopamine Janus nanostructure provided photothermal activity and accurate NIR-light-controlled NO release for a strong antibacterial effect at 200 mg/mL via cellular membrane damage, the leakage of intracellular components, and interference with bacterial metabolism by up- or downregulating genes [213].

Since varying the nanocomposite ratios within a platform plays an essential part in antibacterial efficacy, attention has been paid to this controllable aspect. In the case of BDPNO@PEG-b-PCL micelles, the efficiency changed with the feeding ratios of PEG-b-PCL and BDP-NO. For instance, only NP-4 or NP-5 (1–5 levels) under NIR resulted in evident structural changes, fissures in bacterial membranes, and subsequent cytoplasmic outflow. BDP-NO nanoparticles produced a controllable antibacterial effect via NO release, PTT, or synergistic NO-PTT. Notably, nanoplatforms with NIR-responsive NO generation and PTT, besides promoting NO penetration into the bacterial cell upon PTT-induced bacterial wall damage, could dissolve and remove mature biofilms and, through the released NO, modulate the inflammatory immune response to reduce tissue damage. Therefore, the synergism of NO–PTT proved its antibacterial efficiency in MRSA-infected skin wound models [182].

#### 4.2.3. Antibacterial Photothermal Therapy (PTT)–Chemodynamic Therapy (CDT)

Chemo-photothermal antibacterial therapy combines PTT with chemical drugs such as metal ions and antibiotics. In this respect, temperature-responsive nanostructures are used as carriers for effective antibacterial therapy if they possess satisfactory biocompatibility, encapsulate antibiotics entirely and securely, combine easily as nanoplatforms, release the encapsulated antibiotics quickly under NIR, and synergise PTT-CDT antibacterial activity. The PTT-antibiotic mechanism of action (MOA) interferes with the integrity of bacterial membranes, both thermally and chemically. PTT-CDT synergism may positively influence the outcomes of phototherapy in two ways:

(1) PTT is assisted by CDT to reduce the drug dose, side effects, and drug resistance, and

(2) CDT is combined with PTT to reduce the therapy time and thereby protect normal cells.

Several solutions were proposed based on incorporating antibacterial drugs or nanoenzymes.

The synergetic effects between zeolite imidazole framework-8 and humic acid (HA) (ZIF-8) (HuA@ZIF-8) under NIR light promoted the controlled release of Zn^2+^ ions with antimicrobial activity against *S. aureus* and *E. coli*. ZIF-8 acts as a pH-sensitive vehicle for drug delivery in antibacterial applications. Interestingly, ZIF-8 could be degraded in bacteria-infected areas because of the acidic environment; therefore, it can be incorporated with antibiotics into NIR/pH dual-stimuli-responsive nanoplatforms for the controlled release of an antibacterial drug [214]. Real-time antibacterial drug monitoring was enabled by IMP/IR780@TRN nanospheres, comprising imipenem (IMP, a broad-spectrum antibiotic) and IR780 (a photosensitiser molecule. The NIR-laser-controllable release of IMP at the infection site induced cell wall formation inhibition, while PTT-induced damage to the bacterial membrane effectively killed *E. coli* and *MRSA* [215]. One excellent strategy is to use b-lactam antibiotics to destroy L-form (β-lactam-antibiotic-resistant bacteria) cell walls before PTT: it starts with the disruption of the bacterial cell wall by amoxicillin (AMO) and is followed by PTT. Once such nanoplatform is Pd–Cu/AMO@ZIF-8, or PCAZ, incorporating ZIF-8 loaded with amoxicillin (AMO), which showed significant antibacterial effects in vitro and in vivo (in vitro inhibition rates of *S. aureus* and *P. aeruginosa* were 99.8% and 99.1%, respectively) and destroyed many biofilms under NIR. The significantly decreased infiltration of inflammatory cells, the intact epidermis, and fewer fibrous cells indicated progressive wound healing, facilitated by accelerated drug release in the wounds’ acidic environment under NIR [87]. The strong NIR absorbance associated with excellent particle-size uniformity, such as in CuS@Van and cCuS@Van nanoplatforms, allowed more than a simple synergistic interaction. The low-cost, easy-to-prepare, biocompatible nanocomposites comprising CuS NPs and vancomycin (Van) presented a tri-modal photokilling solution as a potential vancomycin-resistant pathogenic bacterial ablation method. Based on CDT-PTT-PDT, the nanoplatforms exhibited effective antibacterial capability and caused rapid infection regression in vitro and in vivo [216].

CDT reagents with a potent catalytic character have been applied in infected wound treatments [195], while nanoenzyme-based chemodynamic therapy (CDT) has shown tremendous potential in treating bacterial infections. However, the CDT antibacterial efficacy is severely limited by the catalytic activity of nanoenzymes or the infection microenvironment conditions, such as insufficient hydrogen peroxide and over-expressed glutathione (GSH). Therefore, synergistic combinations are considered anti-infective therapies [217]. For instance, Zhu et al. [218] described cationic chitosan@ ruthenium dioxide hybrid nanoenzymes for photothermal-therapy-enhanced CDT in multidrug-resistant bacterial infections. CHFH (CuS_NPs_-HA-Fe^3+^-EDTA hydrogel) is a bacteria-triggered multifunctional hydrogel constructed for low-temperature photothermal sterilisation and high-efficiency integrated localised chemodynamic therapy (L-CDT). CuSNPs act as photothermal agents for low-temperature photothermal therapy (LT-PTT). The network of hyaluronic acid (HA) and Fe^3+^-EDTA complexes allow bacteria to accumulate on the surface, where secreted hyaluronidase decomposes HA and releases Fe^3+^ to be reduced to Fe^2+^ in the bacterial microenvironment. Integrating short-range L-CDT and LT-PTT for sterilisation improved the antibacterial efficiency while minimising damage to normal tissues. Furthermore, CHFH in a band-aid has a potential clinical application, effectively promoting the *S. aureus*-infected wound healing process in vivo [219]. Indocyanine green photosensitisers and AgNPs were simultaneously introduced onto the surface of MoS_2_ nanosheets for combined chemo/photothermal/photodynamic triple-mode antibacterial therapy [160]. Another versatile hybrid nanoenzyme was constructed by grafting ultrasmall CuO_2_ nanodots onto hydrangea-like MoS_2_ nanocarriers for synergistic PTT/CDT dual-mode antibacterial therapy. The MoS_2_/CuO_2_ nanoenzymes’ evaluation revealed a better antibacterial MOA through improved photonic hyperthermia catalytic activity and ROS-based effects by redox. The induced disturbed homeostasis was possible due to the co-catalysis-boosted peroxidase-mimicking activity, H_2_O_2_ self-supplying ability, and GSH-depleting property via oxidisation of GSH to GSSG by both Cu^2+^ and Mo^6+^ within the system. A 99% in vitro antibacterial efficacy against *S. aureus* and *E. coli* was reached at 50 μg mL^−1^ and 100 μg mL^−1^, respectively. A possible cause of this difference was the thinner and more porous cell membrane of *S. aureus*. The studies indicated that, in vitro and in vivo, NIR-MoS_2_/CuO_2_ caused severe damage and more intense bacterial collapse and deformation, effectively eliminating *S. aureus* and *E. coli*, thus showing strong PTT/CDT dual-mode synergistic antibacterial properties with good biosafety and nontoxicity. MoS_2_/CuO_2_ nanoenzymes were proven to have good biosafety and nontoxic effects on NIH_3_T_3_ cells and promoted cell growth, thus showing promise for further applications [220]. Developing intelligent nanotherapeutics for antibacterial therapy is supported by the satisfactory in vitro and in vivo bactericidal activity of a novel infection-microenvironment-responsive PTT/CDT synergistic nanoplatform, Cu_1.94_S@MPN, constructed by encapsulating Cu_1.94_S with Fe(Ⅲ)/tannic acid (TA)-based metal–polyphenol network (MPN) nano-shells. The excellent inherent photothermal conversion properties of MPN and continuous Cu(Ⅰ) ion supply via reducing Cu(Ⅱ) with TA achieved self-boosted synergistic PTT/CDT effects with extraordinary photothermally and photothermally enhanced chemodynamic efficiency [221]. The functionalisation of carbon dots (CDs) as a nanoplatform for synergistic antibacterial chemodynamic and photothermal therapy was proposed. A CD/iron oxychloride nanosheet (CD/FeOCl NSs) catalyst with hydrogen peroxide (H_2_O_2_) induced CDT, while the coating of inorganic CDs and NIR organic polyethylene glycol PEG mediated PTT. The developed FeOCl@PEG@CDs NCs employed Fe(II)’s improved selectivity towards ·OH generation. Efficient synergistic CDT/PTT in the FeOCl@PEG@CDs NCs plus H_2_O_2_ and FeOCl@PEG@CDs NCs plus laser groups significantly deformed the bacterial surface and inhibited and killed the bacteria. The good in vitro and in vivo antibacterial results are attributes of the CDT effect of FeOCl combined with the PTT effect of CDs. Moreover, the CDs’ excellent biosafety and high photostability increase the FeOCl@PEG@CDs NC nanocomposite’s potential as an antibacterial agent and wound microenvironment regulator [194]. Significantly, a self-assembled microsphere hydrogel scaffold (SMHS) regulating the diabetic wound microenvironment via synergistic PTT/CDT might play an essential role in the healing process. SMHS+NIR can combine physical (photothermal therapy) and chemical (drug delivery) mechanisms to significantly increase the anti-inflammatory response, angiogenesis, and tissue remodelling simultaneously. Two kinds of hydrogel microspheres with opposite charges were independently prepared for SMHS: chitosan methacryloyl (CS) and hyaluronic acid methacryloyl (HA). The positively charged CS microspheres were loaded with PEGylated black phosphorus (BP) nanosheets, while the negatively charged HA microspheres were loaded with basic fibroblast growth factor (bFGF). The BP nanosheets provide an efficient photothermal response under NIR irradiation and naturally degrade into PO_4_^3−^ or HPO_4_^2−^ in a physiological environment while being tissue-friendly. BP@CS and NIR irradiation may be the main contributors to the bactericidal effect via membrane rupture, as per the investigation of SMHS against Gram-negative *E. coli* and Gram-positive *S. aureus*. Moreover, chitosan provides some antibacterial effects, and BP-PEG can strengthen antimicrobial activity via a photothermal effect. An in vivo study showed several promising outcomes in the group treated with SMHS+NIR: early enhanced local angiogenesis (day 3), better re-epithelialisation (day 14), a typical near-to-normal histological architecture, and an ameliorated inflammatory environment promoted by macrophage polarisation to the M2 subtype [222]. Still, overcoming the PTT-related side effects is crucial. Therefore, a bimetal-doped nanosheet (FeS@Cu_2_O) was proposed. Fabricated via a hydrothermal method, it integrates photothermal, photodynamic, and chemodynamic properties. FeS and Cu_2_O are considered ideal photothermal agents individually due to electron–hole generation and relaxation under NIR. However, FeS@Cu_2_O releases Fe2+ and Cu+ and has a superior photocatalytic ability for PDT and CDT. Therefore, the nanocomposites induce efficient antibacterial effects against *E. coli* and *S. aureus* via local hyperthermia and endogenous ROS as a versatile multimodal synergistic therapy for sterilisation [223].

#### 4.2.4. Other Combinations

Sonodynamic therapy (SDT) combines acoustic sensitisers and low-intensity ultrasound to activate the sensitisers, focus the ultrasound energy deep into the infected tissue, produce mechanical damage via cavitation and sonoporation [224], and induce cytotoxic ROS via sonoluminescence for a bactericidal effect. Moreover, sonoporation can increase cell membrane permeability, thus increasing transmembrane drug transport [225]. Since sonoluminescence at this stage is difficult to regulate and causes toxic or other side effects on healthy tissues, combining SDT with PTT may synergise the therapeutic effects and diminish damage to normal tissues. Most photosensitive agents have the potential to act as acoustic sensitisers. For instance, silver peroxide NPs (Ag_2_O_2_ NPs) are a potential antibacterial drug due to their good photothermal and acoustic sensitiser properties, and they can produce ROS and penetrate deeper into infected tissues with more treatment specificity [226]. To improve the antimicrobial specificity, PTT combined with immunotherapy as an antibacterial option was studied using advanced immunoconjugates. These are adjuvants or antigens altered to antibacterial agents to facilitate complex immune responses via bacterial toxicity and bacterial antigen recognition [227]. One example is the nano-neuro-immune blockers (NNIBs) obtained by adding an immune-escape outer membrane to Au nanocages’ surfaces (AuNCs) [228]. NNIBs have specificity towards the toxins of *S. pyogenes*, neutralise streptolysin S (SLS), unblock neutrophil production, and enhance the host’s immune response to the bacterial infection. Additionally, AuNCs’ good photothermal activity effectively induces an acute inflammatory response with positive feedback on the immune response. Furthermore, a photoexcited hydroxyapatite (Hap)/nitrogen-doped carbon dot (NCDS)-modified GO heterojunction film (GO/NCD/Hap) improved the photocatalysis and photothermal effects with beneficial consequences on tissues, such as in vivo vascular injury repair [229]. The electron transfer synergises PDT and PTT through immune therapy to better treat bacterial infections [230]. Therefore, the combination of PTT and SDT or immunotherapy is anticipated to have more application prospects due to its minimally invasive nature and higher antibacterial effect.

In conclusion, photothermal therapy combined with other photodynamic treatments can play a synergistic role in improving antibacterial efficiency, shortening the antibacterial time, and reducing the side effects of different methods on the human body when used alone, making this a promising therapeutic direction. One important application is wound healing, particularly diabetic and slow-healing surgical wounds, which are extremely susceptible to drug-resistant bacterial infections.

### 4.3. Cutaneous Wounds

The worldwide wound care marketplace, valued at USD 20.8 billion in 2022, is anticipated to reach a compound annual growth rate (CAGR) of 5.4% by 2027 (USD 27.2 billion) [175]. Wound healing is a complex process involving inflammation, proliferation, epithelialisation, and remodelling at haemostasis [26], and it is hindered by bacterial infections, especially in chronic wounds, such as diabetic foot ulcers, non-healing surgical wounds, or peripheral vascular diseases [231,232,233] (Figure 3). Therefore, efforts are focused on solutions that can protect wounds by defeating antibiotic-resistant strains and promoting the healing of either superficial (wavelengths below 400 nm) or deep (400 to 850 nm) cutaneous wounds.

One functionalised water-soluble photothermal agent modified with quaternary ammonium salts (RT-MN) based on electrostatic adsorption was proposed for photothermal antibacterial treatment. The principle of electrostatic interaction facilitates RT-MN molecules’ binding to bacterial membranes to assist in photothermal antibacterial treatment. Since RT-MN is positively charged, whereas *MRSA* and *E. coli* are negatively charged, it can bind to bacteria through electrostatic adsorption. Moreover, besides its exceptional photothermal conversion ability (irradiation with lasers of 808 nm, 150 µM optimal concentration), RT-MN possesses a selective bactericidal effect at high temperatures. RT-MN +NIR destroyed the bacterial membrane and effectively inhibited the growth of *MRSA* and *E. coli*. Subsequently, the in vivo antibacterial ability was successfully demonstrated in an *MRSA*-infected mouse skin wound model. The good biocompatibility of RT-MN combined with NIR irradiation successfully reduced the sizes of infected wounds and facilitated healing via an anti-inflammatory response and increased collagen secretion [234]. Similarly, NIR-irradiated Au@CD-based membranes effectively eradicated bacteria at the wound site, reduced the risk of bacterial infection, suppressed inflammation, and improved collagen deposition and angiogenesis, facilitating wound closure via a photothermal antimicrobial effect. The healing platform comprised a PVA membrane, which embedded AuNPs and N,S-CDs via electrospinning. Using CDs as surface decorations conferred improved photothermal conversion efficiency, photostability, and biocompatibility to Au@CD compared to the parent AuNPs. The membranes presented excellent biocompatibility and photothermal antimicrobial activity against S. *aureus* and *E. coli* (99% inactivation of both pathogens under NIR irradiation) in vitro and in vivo [235].

As previously mentioned, wound-dressing hydrogels have attracted much attention due to their interconnected microporous networks, which can maintain a humid microenvironment and promote the absorption of wound exudate and oxygen transmission. Hydrogels effectively improved the permeability of bacterial membranes, ruptured the bacterial membrane, and induced oxidative stress and significant protein leakage and, thus, bacterial death. Importantly, due to its negligible side effects, this system has great clinical potential for sterilisation through the combination of PTT and PDT. Therefore, nanocomposite hydrogels with antibacterial and antioxidant capabilities have great application potential in treating infected skin wounds. For instance, introducing anils as antibacterial agents into polyvinyl alcohol (PVA) hydrogel is a good choice to achieve a rapid antibacterial therapeutic effect. An antibacterial platform (DPVA hydrogel), mainly derived from the photothermal effect of N-(2,4-dihydroxybenzylidene)-4-aminophenol (DOA), was tested for the efficient and rapid treatment of drug-resistant bacterial infections in skin wounds. An excited-state proton transfer (ESIPT) process with nonradiative transitions was utilised to promote the photothermal effect and increase the local temperature to 55 °C within 10 s under irradiation. In Vitro evaluations showed a broad-spectrum antibacterial ability against *S. aureus* and *E. coli* (antibacterial rate 99%) and methicillin-resistant *S. aureus* and Enteroinvasive *E. coli* (about 1% bacterial survival rate). In vivo wound healing in mice showed that the DPVA hydrogel could effectively cure *MRSA-*induced whole-layer wound infections within 100 s of irradiation, opening a new way to develop antibacterial dressings with a rapid response and convenient fabrication [54]. Also notable is the 98% wound-healing rate with complete re-epithelisation after a 10-day treatment with MXenes@PVA plus NIR. The MXenes@PVA hydrogel was effective against *S. aureus* and presented high toughness, anisotropy, and antimicrobial properties, thus making it a promising antibacterial dressing for wound healing.

Ti_3_C_2_Tx/Ag_3_PO_4_, Ti_3_C_2_Tx/MoS_2_, Ti_3_C_2_Tx/Bi_2_S_3_, Ti_3_C_2_Tx/Ag, and Nb_2_C are other composites that also presented potential in in vivo wound-healing capabilities [99]. Furthermore, BSA@MPN, produced by Fe^3+^/EGCG-based self-assembly using BSA as the nanoreactor and colloidal stabiliser, possessed excellent photothermal-related bactericidal activity and macrophage M1-to-M2 phenotypic-conversion-related anti-inflammatory effects due to the photothermal properties and pH-responsive degradability of Fe^3+^/EGCG-based MPN. Next to biocompatibility and biosafety properties, in vitro and in vivo antibacterial and anti-inflammatory effects are imperative to wound healing [174].

Since wound healing involves wound management, developing an ideal hydrogel-based dressing is essential. However, producing a smart and dynamic hydrogel to adjust to the wound-healing process is still challenging. The dressing needs to have complex properties to facilitate healing [236,237]. Both elasticity and antibacterial properties were obtained for a GO hybrid hydrogel scaffold prepared by injecting a benzaldehyde- and cyanoacetate-group-functionalised dextran solution containing GO into a pool of histidine. This scaffold also allowed enhanced cargo due to the thermosensitive C double-bond breaking under NIR light and at a high temperature [238]. Since GO possesses obvious photothermal behaviour, it was also considered for loading an electrospun hyaluronic acid membrane. The proposed wound dressing loaded with GO and ciprofloxacin exerted a bactericidal effect based on the synergistic action of the antibiotic and NIR-mediated hyperthermia [239,240]. A similar synergistic effect was observed in another NIR-responsive rGO hybrid cryogel with excellent properties: the rGO photothermal effect allowed cryogel heating and subsequent swelling, followed by the encapsulated drug’s release [241]. The elevated local temperature by the photosensitiser graphene denatures the microbial proteins, which decreases the viability of pathogenic microorganisms, *MRSA* included [240]. Furthermore, combining PTT and PDT was considered. Chitosan-oligosaccharide-functionalised graphene quantum dots (GQDs-COS) with short-term exposure to 450 nm visible light [242] and a nanocomposite poly(vinylidene) fluoride membrane with TiO_2_ NPs as the outer shell and NaYF4:Yb,Tm nanorods as the core doped with nanosized GO as a photothermal agent [173] were proposed. The suggested MOA is complex PTT-PDT synergism initiated by PTT, completed by PDT, and manifested by increased electrostatic attraction between the NPs and the bacteria, the irreversible disintegration of bacterial cytomembranes, leaked cytoplasm, oxidised vital subcellular targets, disturbed bacterial homeostasis, and bacterial death. Therefore, the GO hybrid hydrogel scaffolds could be used as multifunctional wound dressings because of their photothermal antibacterial, adjustable mechanical, and angiogenesis-promoting properties. In vivo (5 days of treatment on mice) results suggested that NIR-laser-assisted MoS_2_/CuO_2_ nanoenzymes effectively eliminated *S. aureus* infection by a PTT/CDT dual-mode synergistic antibacterial effect, finally achieving adequate wound healing. The results confirmed the potential of MoS_2_/CuO_2_ as an effective PTT agent for future biomedical applications [220]. Chang et al. demonstrated successful PPT-PDT synergism as a potential modality in treating infected skin wounds. The proposed functional wound dressing combined an enzyme-crosslinked hyaluronic acid–tyramine (HT) hydrogel and antioxidant and photothermal AgNPs capped with tannic acid (AgNPs@TA). The natural antioxidant tannic acid (TA) acted as a reducing and stabilising agent to facilitate synthesis. The HTA hydrogel is biocompatible and easy to use, while AgNPs@TA significantly enhanced the photothermal, antioxidant, antibacterial, adhesive, and haemostatic abilities of the resulting nanocomposite. In vivo studies on *S. aureus* and *E. coli* co-infected mouse skin wound models showed that HTA0.4 (containing 0.4 mg/mL AgNPs@TA) hydrogel combined with NIR radiation greatly reduced inflammation, helped angiogenesis, and enhanced the healing process. Therefore, the antibacterial and antioxidant AgNPs@TA is a promising wound dressing [205]. The DPVA hydrogel was also evaluated for its efficient photothermal antibacterial effects with potential application as a wound dressing for infected wounds. In Vitro and in vivo studies showed excellent antibacterial effects against methicillin-resistant *S. aureus* and *E. coli*, explained by the completely disrupted bacterial structure due to the generated PTT-induced hyperthermia. Notably, new epidermal tissue formation indicated the healing process [54]. Angiogenesis and collagen deposition, as healing steps, were facilitated by a biocompatible adhesive nanocatalytic hydrogel with a polyvinyl alcohol (PVA) scaffold, MXene/CuS bio-heterojunctions, and polydopamine (PDA). MXene/CuS’s photothermal effect and NIR-light-generated oxygen and hydroxyl radicals explained the good antioxidant and antibacterial capacities, thus the good in vivo skin regeneration ability through bactericidal, angiogenesis, and collagen deposition promotion. The proposed approach requires further research for nanocatalysed hydrogels with an infection-microenvironment-induced response to treat infected wounds through enhanced phototherapeutic effects [206,243]. Ag-NP-incorporated quaternised chitin nanocomposite was constructed by an in situ synthesis method for application in biofilm-infected wound treatment for the first time. Ag^+^ was reduced in situ to Ag NPs stabilised by catechol-functionalised quaternised chitin (DQC) micelles to form DQCA in a green approach, without extra reductant or UV irradiation. The rationally designed DQCA would be endowed with bacterial targeting, sterilisation effects of cationic groups and Ag NPs, and superior combined photothermal bactericidal and antibiofilm activities. Furthermore, the DQCA solution was injected into a full-thickness *S. aureus* biofilm-infected wound in a mouse model to demonstrate the application prospects for wound healing. This study provided a new multifunctional silver/polysaccharide candidate for treating biofilm-infected wounds [81]. In Vitro and in vivo studies presented a porphyrin-based covalent organic framework, TP-PorCON@BNN6, as a triple antibacterial model, with good biocompatibility, negligible toxicity, and multifunctional biological activity. It promoted bacterial apoptosis by producing ROS, causing hyperthermia, and releasing NO via the synergistic effect of PDT, PTT, and GT. The in vivo antibacterial activity was quantified on S. aureus-infected chronic wounds receiving different treatments (PBS, PDT, PDT + PTT, and PDT + PTT + GT). The accelerated healing of infected wounds by simultaneously reducing oxidative stress, regulating inflammatory factors, accelerating collagen deposition, and promoting angiogenesis was observed [145]. Other multifunctional antibacterial nanoplatforms could employ PTT, PDT, and SDT for guided therapy and rapid healing. For instance, an AIE-Tei@AB NV was designed as a laser-activated “nanobomb” for the multimodal theranostics of drug-resistant bacterial infections. It comprises lipid nanovesicles from the self-assembled aggregation-induced emission (AIE) nanosphere (AIE-PEG1000 NPs) with near-infrared region II (NIR-II) fluorescence emissive, photothermal, and photodynamic properties. Furthermore, the nanobomb combined the excellent pharmacological properties of rapidly released Tei during bubble generation and the NV disintegration capacity. Therefore, in vivo experiments with high-performance NIR-II fluorescence, infrared thermal, and ultrasound imaging of multidrug-resistant bacteria-infected foci validated the broad-spectrum eradication of clinically isolated *MRSA*, MDR *E. coli*, and MDR Pseudomonas aeruginosa and the rapid healing of infected wounds upon intravenous administration of AIE-Tei@AB NVs followed by 660 nm laser stimulation. This multimodal imaging-guided synergistic therapeutic strategy can be extended for the theragnostics of superbugs [244].

Ultimately, an intelligent wound dressing is intended to regulate the microenvironment by attenuating inflammation responses and promoting re-epithelisation, granulation formulation, angiogenesis, and collagen deposition. The literature strongly suggests that NIR exposure has multifunctional effects and holds enormous potential for wound therapy.

In conclusion, the need for two-dimensional nanomaterials as promising candidates for wound healing brings photothermal agents to the forefront. They can provide an efficient photothermal response under NIR irradiation and naturally degrade in a physiological environment, being completely harmless to the surrounding tissues.

## 5. Conclusions, Challenges, and Perspectives

This review presents the last five years of nano-antibacterial technology and describes how nanotechnological progress has introduced many new nanomaterials as PTT agents (PTAs), showing good antibacterial activity in controlled nanosystems. The microbial aetiology and pathogenesis are complex, and the imbalanced microenvironment and homeostasis cause the most related health problems. Acidic bacterial metabolites may induce local tissue damage, and the lipopolysaccharides (LPS) released by bacterial biofilms trigger inflammation and long-term issues. Emerging nanotechnology can bring antibacterial treatment to a new level. Many photodynamic nanomaterials are potent bactericidal agents against common cutaneous microbial pathogens (e.g., *S. aureus*, *E. coli*, and *P. aeruginosa*) and inhibit initial bacterial colonisation or biofilm formation. This review presents various methods to ensure targeted PTAs’ actions via their bacterial specificity to improve their concentration in the targeted infected area. However, single antibacterial therapy encounters several challenges, from complex technological processes to pharmacological and toxicological effects on local and distant infected or healthy tissues to cost efficiency. Since technological and safety concerns are intertwined, the research and development of antibacterial materials and modalities should change with the type and concentration of nanomaterials based on the distinctive targeted environments and the most suitable technological and financial combinations. Based on the characteristics of the infected environment and the therapeutic needs imposed by various pathogens’ and materials’ responses to the infected microenvironment, the design of antibacterial materials should follow consistent standards, such as:Close-to-zero toxicity to the human body,Close match with the unique needs of different situations (e.g., re-epithelisation, anti-inflammatory effect),High specificity and sufficient penetration of the targeted lesions,The ability of the intelligent therapeutic multimodality to be incorporated into smart and dynamic nanoplatforms, such as hydrogels, to adjust to the wound-healing process,Low cost and easy to mass produce.

The main challenges may become the foremost opportunities for new functionalised, highly biocompatible PTAs with increased photothermal efficiency for future clinically applicable nanosystems.

Therefore, new nanotechnology-based approaches have considered multimodal solutions to combine PTAs with other materials to improve the antibacterial efficiency and reduce unwanted side effects. The technological outcomes presented here explain the recent applications of PTT to sterilisation by removing bacterial biofilms from different medical surfaces and implants and to the rapid healing of infected wounds via the molecular implications of PTT alone or in synergistic combinations. Despite the recent in vivo progress in PTA-based antimicrobial agents, the transition of these materials from benchtop to bedside still needs to be improved. For instance, in the case of PTAs combined with antibiotics, there is a risk of unpredictable drug distribution and suboptimal local concentration, which may limit its therapeutic effect. This example also explains the need for protocol standardisation as one crucial issue that needs to be addressed in antimicrobial PTAs. However, this goal is challenging to accomplish since there are multiple variables to control simultaneously. It is not easy to consolidate the wide variety of nanomaterials employed at various concentrations, with different power densities, and for various NIR exposure times for the best efficacy and highest safety to the tissues.

Consequently, it is not easy to compare antimicrobial potency between different PTAs or to enable better translation to the clinical setting. Notably, only a limited number of studies address the biocompatibility of these photothermal nanomaterials and their stability issues in the tissue environment. Safety concerns stem from exposure to nanoagents, which could evade or enhance the body’s immune mechanisms, and from biohazards released into the environment because of in vitro testing. Therefore, rigorous in vivo testing and methods to improve biocompatibility (e.g., biosynthesis, optimisation of antibacterial efficiency) are subjects of further research. However, most nanoproducts have complex synthetic paths and high costs at the present stage, which hinder their mass production. Furthermore, translating nanoproducts into clinical applications requires clinical trials, which necessitates improvements in health and safety. New strategies to explore these products may include combining emerging fields, such as biosensors, for intelligent, on-demand, and precise strategies (e.g., pH-, temperature-, or enzymatic-activity-responsive nanodevices, mild and low PTT, in addition to NIR PTT, MOF-PTT, COF-PTT, or MN-coupled PTT nanoplatforms for antimicrobial delivery combined with photodynamic therapy and wound healing).

Moreover, future studies will focus on biosafety to address PTAs’ interactions with the immune system, their long-term biosafety, the assessment of potential degradation products, and the effects of the undesirable diffusion of the nanomaterials. Research could focus on more systematic investigations of the PTAs’ antibacterial mechanisms with the help of in silico prediction models (e.g., the efficacy against different types of bacteria biofilms, detailed antibiofilm PTAs’ mechanisms of action). Eventually, well-structured studies will investigate the pharmacological aspects of dynamics, kinetics, and toxicology for detailed profiles of PTAs. Aspects include administration routes, biodistribution, nanoparticle metabolism and excretion (clearance), off-target effects of NIR irradiation, cytotoxicity, and interaction with the commensal microbiota.

Finally, the intense exploitation of natural compounds to decrease manufacturing costs and the negative impact on the environment could be considered. Therefore, nanotechnology must take an essential step towards the large-scale production of synthesised nanomaterials in laboratories for their translation into widespread clinical therapeutic devices. Hence, resolving the shortcomings of PTT and PTT-derived multimodal therapies is essential, as they provide substantial tissue penetration and synergistic bactericidal prospects. For instance, short- and long-term biological limitations required functionalising PTAs for specific bacteria and moving towards lower-energy or active photothermal materials for multimodal synergistic-based therapies to minimise side effects and maintain low costs (e.g., PTT-PDT, PTT-CDT, PTT-photocatalytic, PTT-immunotherapy, and PTT-catalytic). Successful design, characterisation, and transition to the clinical setting will only be possible due to a comparative, interdisciplinary approach involving clinicians, pharmacists, engineers, microbiologists, chemists, and industrial partners.

## Figures and Tables

**Figure 1 ijms-24-09375-f001:**
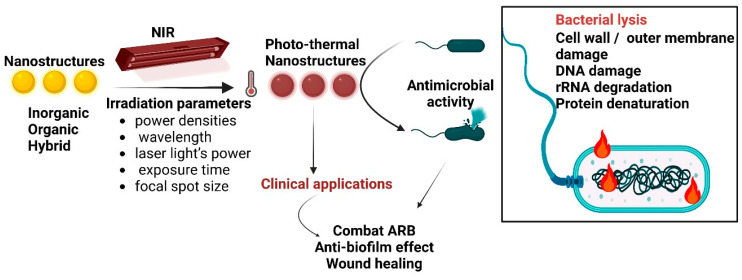
Schematic photothermal antimicrobial mechanism of PTA nanostructures. PTA nanostructures (with different structures and exposed to different parameters of irradiation) act by heat-induced mechanisms, culminating in bacterial lysis; ARB—antibiotic-resistant bacteria.

**Figure 2 ijms-24-09375-f002:**
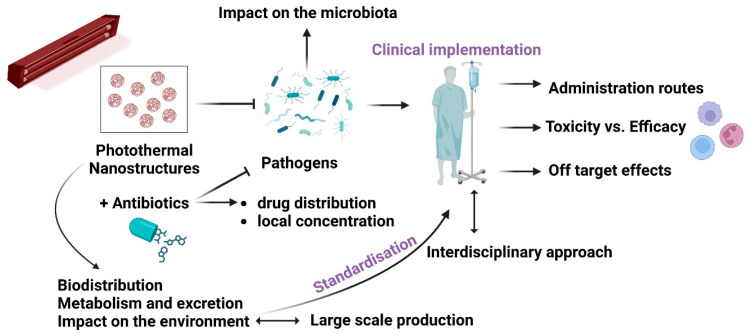
PTA nanostructures—from bench to bedside: the main challenges to be tackled.

**Figure 3 ijms-24-09375-f003:**
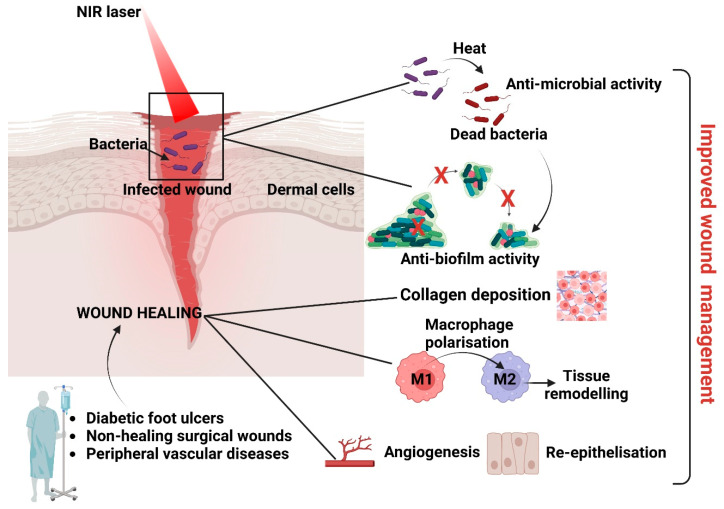
Impact of PTA nanostructures on wound healing. Photothermal nanostructures can impact wound healing by targeting invading microbes and preventing biofilm formation and by promoting re-epithelisation, angiogenesis, collagen deposition, and activation of M2 macrophages (involved in tissue remodelling).

**Table 2 ijms-24-09375-t002:** Representative C-based PTA nanomaterials for antibacterial activity.

	Matrix/ Material	Light (nm) and Intensity	Temperature Reached	Antibacterial Mechanism	In Vitro Biological Performance		Ref.
					Type of Bacteria	Efficacity	
**Carbon-based nanomaterials**	rGO/AuNP-nanostars	808 nm; 3.0 W/cm^2^	73.5 °C	PTT	*S. aureus* *E. coli*	100%	[124]
	MWNT/DTTC	808 nm; 1.0 W/cm^2^	92 °C, 120 °C	PTT	*P. aeruginosa*	77–100%	[125]
	GO/Ag	808 nm; 1.5 W/cm^2^	24.6 °C	PTT and Ag^+^ release	*MDR E. coli*	~96%	[127]
	rGO/Ag	808 nm; 0.30 W/cm^2^	Higher with ~20 °C	PTT and Ag^+^ release	*E. coli* *K. pneumonia*	100%	[126]
	AgNPs PVP@rGO	Visible light	-	PTT and Ag^+^ release and physical wall demolition	*E. coli*	Effective	[129]
	Fe_3_O_4_@GO-QCS	808 nm; 3.0 W/cm^2^	≥50 °C	Bacteria capture and PTT and Magnetic Recycling	*S. aureus* *E. coli*	~100%	[130]
	Fe_3_O_4_-CNT-PNIPAM	808 nm; 3.0 W/cm^2^	-	Bacteria capture and PTT and Magnetic Recycling	*S. aureus* *E. coli*	~100%	[131]

Abbreviations: AuNP-nanostar—gold nanostar; MWNT—multiwalled carbon nanotubes; DTTC—3,3′-diethylthiatricarbocyanine fluorophores; AgNPs PVP@rGO—polyvinylpyrrolidone-functionalised silver nanoparticles combined with reduced graphene oxide; GO-QCS—quaternised-chitosan-anchored graphene oxide; Fe_3_O_4_-CNT-PNIPAM—poly(N-isopropylacrylamide) chemically grown onto the surface of carbon nanotube (CNT)-Fe_3_O_4_.

## Data Availability

No new data were created.

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
