# Peer review of "Trends in Photothermal Nanostructures for Antimicrobial Applications"

_ijms, 2023, doi:10.3390/ijms24119375_

Round 1
Reviewer 1 Report
In the following review, “Trends in photothermal nanostructures for antimicrobial applications”, the authors claimed to have reviewed the recent progress on photothermally active nanostructures, including plasmonic metals, semiconductor, carbon-based, and organic photothermal polymers, and also highlighted the antibacterial mechanisms of action, including anti-multidrug resistant bacteria and biofilms removal processes.
1) A review article aims at achieving THREE essential objectives:
i) providing a complete, structured and systematic summarization on the related key aspects. This means that the authors will summarize those in many figures and tables;
ii) presenting new discoveries from the authors’ own knowledge synthesis based on existing literature results. This means that the authors will provide important and synthesized new knowledge that are not included in those articles in the literature; and
iii) outlining detailed views on future research directions and perspectives.
Objectives 1) has been reasonably achieved, indeed not many recent reviews are present in the literature on the selected topic, therefore, the summarized knowledge may offer valuable information to the interested readers. However, this review lack proper graphical demonstration which can be enhanced by including some figures from corresponding literature reports
Objective ii) has also been decently achieved through much of the literature and the authors' own critical thinking via comparing/benchmarking the literature to offer new knowledge.
Objective iii) has been partially achieved, this section can be further enhanced by including more past achievements and offering solutions to current challenges. For further comments (see below).
At several places, minor typographical errors have been found.
At some places, the paragraphs are too large
Cite few more relevent references such as,
"Graphene based metal and metal oxide nanocomposites: synthesis, properties and their applications." Journal of Materials Chemistry A 3.37 (2015): 18753-18808.
Therefore, the authors are advised to perform some minor revisions to make the review paper suitable for publication.
See the report
Author Response
We thank the reviewer for the time to provide feedback and for the help to improve the manuscripts. The modifications in the new version of the manuscript were marked with red. Please find the answers to the comments as follow.
Comments: In the following review, “Trends in photothermal nanostructures for antimicrobial applications”, the authors claimed to have reviewed the recent progress on photothermally active nanostructures, including plasmonic metals, semiconductor, carbon-based, and organic photothermal polymers, and also highlighted the antibacterial mechanisms of action, including anti-multidrug resistant bacteria and biofilms removal processes.
1) A review article aims at achieving THREE essential objectives:
- i) providing a complete, structured and systematic summarization on the related key aspects. This means that the authors will summarize those in many figures and tables;
- ii) presenting new discoveries from the authors’ own knowledge synthesis based on existing literature results. This means that the authors will provide important and synthesized new knowledge that are not included in those articles in the literature; and
iii) outlining detailed views on future research directions and perspectives.
Objectives 1) has been reasonably achieved, indeed not many recent reviews are present in the literature on the selected topic, therefore, the summarized knowledge may offer valuable information to the interested readers. However, this review lack proper graphical demonstration which can be enhanced by including some figures from corresponding literature reports.
Response: Thank you for your comments and suggestions, we add one more figure and we make the conclusion part more “pointed”.
Objective ii) has also been decently achieved through much of the literature and the authors' own critical thinking via comparing/benchmarking the literature to offer new knowledge.
Objective iii) has been partially achieved, this section can be further enhanced by including more past achievements and offering solutions to current challenges. For further comments (see below).
Response: We improve the conclusion part underlining more the challenges as well as the trends in the field.
At several places, minor typographical errors have been found.
Response: Thank you, we checked and amended accordingly.
At some places, the paragraphs are too large
Response: Thank you, we checked and amended accordingly.
Cite few more relevant references such as,
"Graphene based metal and metal oxide nanocomposites: synthesis, properties and their applications." Journal of Materials Chemistry A 3.37 (2015): 18753-18808.
Response: Thank you, we checked and amended accordingly.
Reviewer 2 Report
The authors did outstanding work on reviewing the current photothermal antibacterial nanomaterials and their recent application in this field. They represented the recent material in organized classification and also the applications.
The work is really sufficient and interesting to the people who want to start work in the photothermal antimicrobial field.
In my review, I have follow suggestion after carefully reading:
1. Line 80, PTT would be PDT or it is photothermal therapy, there is no word indicate it.
2. Since there is a lot of abbreviation in the paper, I suggest them to organize the abbreviation and put them at the beginning or end of the manuscript.
3.Line 230, us there any reference show the compounds based PTA? If my understanding is correct, the table 1 is the example for this section. Then why not put some sentence mention the table in this paragraph since the tale is not connected to the paragraph
4. Similar to table 2, suggest mentioning the table in their connected paragraph. Meanwhile, how about separating table 2 to two tables? Since they mention both C-based and organic based, that may be reasonable to separate them and put them under the correlated paragraph.
Author Response
We thank the reviewer for the time spend to provide feedback and for the valuable comments.. The modifications in the new version of the manuscript were marked with red. Please find the answers to the comments as follow.
- Line 80, PTT would be PDT or it is photothermal therapy, there is no word indicate it.
Response: Thank you, we checked and amended accordingly, it is PTT.
- Since there is a lot of abbreviation in the paper, I suggest them to organize the abbreviation and put them at the beginning or end of the manuscript.
Response: Thank you, we checked and added on list at the end of the manuscript.
3.Line 230, us there any reference show the compounds based PTA? If my understanding is correct, the table 1 is the example for this section. Then why not put some sentence mention the table in this paragraph since the tale is not connected to the paragraph
Response: Thank you, we checked and amended accordingly. There are short indications in text to refer the reader to Tables 1, 2 and 3.
- Similar to table 2, suggest mentioning the table in their connected paragraph. Meanwhile, how about separating table 2 to two tables? Since they mention both C-based and organic based, that may be reasonable to separate them and put them under the correlated paragraph.
Response: Thank you, we checked and amended accordingly. There are short indications in text to refer the reader to Tables 1, 2 and 3. We also split Table 2.
Reviewer 3 Report
The authors provide a rather detailed and comprehensive review on the photothermal nanostructures for antimicrobial applications. The review is topical, although it is mainly focused on the state of the art rather than on the trends in this topic. I believe some relatively minor issues can be addressed.
- I suggest changing the title to "Review on photothermal nanostructures for antimicrobial applications" or simply to "Photothermal nanostructures for antimicrobial applications" as this is main layer of discussion in the review with the trends as one of the sections (Section 5). In my opinion, this will make the audience even broaded that in the current version.
- I suggest summing up the keywords, databases, and time interval for the literature search for the review
- Figure 1, in my opinion, is too simplified and better suits as a kind of graphical abstract. Probably, such a figure may be either omitted or, contrariwise, replaced with a more detailed figure showing the trends and mani development areas in the selected review topic.
- I do not understand, why the common abbreviation AuNP for all gold nanoparticles is used especially for nanostars (Table 2, line 526).
- I suggest more graphical illustrations would be welcome for this review as the topics are complex and not always easy to be understood by a novice in the field without an illustration.
- I recommend checking the abbreviations for both tables as they are not well formatted and sometimes show different order (abbreviation ten explanation and vice versa; italics use, etc.).
- The column PTT parameters in Table 1 also could be made more consistent and similar as the information is given in a slightly different way in each row.
- In Table 2, the data in the second column is not power but intensity or flux as it is measured in the units of power per area.
Author Response
We thank the reviewer for the time spend to provide feedback and for the valuable comments. The modifications in the new version of the manuscript were marked with red. Please find the answers to the comments as follow.
The authors provide a rather detailed and comprehensive review on the photothermal nanostructures for antimicrobial applications. The review is topical, although it is mainly focused on the state of the art rather than on the trends in this topic. I believe some relatively minor issues can be addressed.
- I suggest changing the title to "Review on photothermal nanostructures for antimicrobial applications" or simply to "Photothermal nanostructures for antimicrobial applications" as this is main layer of discussion in the review with the trends as one of the sections (Section 5). In my opinion, this will make the audience even broaded that in the current version.
Response: Thank you for suggesting a new title. Kindly allow us to keep the original title. We elaborated on the trends (each section ends with a short conclusion), challenges, and perspectives to highlight the main objectives of the research in the field. Chapter 5 highlights the main directions.
- I suggest summing up the keywords, databases, and time interval for the literature search for the review
Response: Thank you, we mentioned the time interval the recent five years (2019-2023) in text at the start of Chapter 5. We didn’t use any database.
- Figure 1, in my opinion, is too simplified and better suits as a kind of graphical abstract. Probably, such a figure may be either omitted or, contrariwise, replaced with a more detailed figure showing the trends and mani development areas in the selected review topic.
Response: Thank you, we edited the figures accordingly to highlight the main objectives of the review and the concepts described
- I do not understand, why the common abbreviation AuNP for all gold nanoparticles is used especially for nanostars (Table 2, line 526).
Response: Thank you, we changed accordingly: AuNP to AuNP-nanostar (table 2)
- I suggest more graphical illustrations would be welcome for this review as the topics are complex and not always easy to be understood by a novice in the field without an illustration.
Response: Thank you, we included original figures to highlight the main objectives of the review and the concepts described
- I recommend checking the abbreviations for both tables as they are not well formatted and sometimes show different order (abbreviation ten explanation and vice versa; italics use, etc.).
Response: Thank you, we added a list at the end of the manuscript to include the main abbreviations such as bacteria names, or the most common substrates. The manuscript has the names are in full.
- The column PTT parameters in Table 1 also could be made more consistent and similar as the information is given in a slightly different way in each row.
Response: Thank you, we checked and did our best to be consisted, given the fact that the available data were presented in various amount and formats
- In Table 2, the data in the second column is not power but intensity or flux as it is measured in the units of power per area
Response: Thank you, we modified accordingly: “power “changed to “intensity” in Table 2 and 3